# PICNIC accurately predicts condensate-forming proteins regardless of their structural disorder across organisms

Anna Hadarovich[1,2,5], Hari Raj Singh[1,5], Soumyadeep Ghosh [1,2], Maxim Scheremetjew[1,2], Nadia Rostam[1,2,4], Anthony A. Hyman [1,2,4] & Agnes Toth-Petroczy [1,2,3] ✉

Biomolecular condensates are membraneless organelles that can concentrate hundreds of different proteins in cells to operate essential biological functions. However, accurate identification of their components remains challenging and biased towards proteins with high structural disorder content with focus on self-phase separating (driver) proteins. Here, we present a machine learning algorithm, PICNIC (Proteins Involved in CoNdensates In Cells) to classify proteins that localize to biomolecular condensates regardless of their role in condensate formation. PICNIC successfully predicts condensate members by learning amino acid patterns in the protein sequence and structure in addition to the intrinsic disorder. Extensive experimental validation of 24 positive predictions *in cellulo* shows an overall ~82% accuracy regardless of the structural disorder content of the tested proteins. While increasing disorder content is associated with organismal complexity, our analysis of 26 species reveals no correlation between predicted condensate proteome content and disorder content across organisms. Overall, we present a machine learning classifier to interrogate condensate components at whole-proteome levels across the tree of life.

Biomolecular condensates are membraneless organelles that can selectively concentrate biomolecules and are typically non-stoichiometric assemblies of thousands of protein molecules and nucleic acids. The role of condensates has been implicated in several fundamental biochemical processes in physiology and disease[1–4]. Functions exerted by condensates include: (i) sequestering molecules and shutting down translation of specific mRNAs[5]; (ii) buffering concentration of proteins[6]; (iii) providing a reservoir of proteins for fast assembly and disassembly of large complexes, such as the nuclear envelope[7].

Proteins can have two major roles in a condensate: drivers (scaffolds) or clients[8,9]. Drivers can induce the formation of condensates and are essential members of condensates. For example, knock-out of

the driver protein can lead to disassembly of a condensate. While a client is recruited to a condensate often via an interaction with a driver protein, and it is neither necessary nor sufficient in driving the condensate formation. Drivers often phase separate in vitro, nevertheless self-phase separation does not guarantee in vivo driver functionality. For most condensate-localizing proteins the client or driver status is unknown, therefore we refer to them as condensate members.

While many proteins can phase-separate in the test-tube and form liquid-like condensates in vitro, studying if they also form condensates in vivo is more challenging. Individual proteins can be labeled using fluorescent tags and imaged for testing droplet formation which exhibit liquid-like properties such as fusion, Oswald ripening, fast dynamics in fluorescent recovery after photobleaching (FRAP)[10,11]

[1]Max Planck Institute of Molecular Cell Biology and Genetics, 01307 Dresden, Germany. [2]Center for Systems Biology Dresden, 01307 Dresden, Germany. [3]Cluster of Excellence Physics of Life, TU Dresden, 01062 Dresden, Germany. [4]Present address: Department of Biology, College of Science, University of Sulaimani, Sulaymaniyah, Iraq. [5]These authors contributed equally: Anna Hadarovich, Hari Raj Singh. ✉e-mail: toth-petroczy@mpi-cbg.de

assay. However, condensates may contain hundreds and even thousands of different proteins. Systematic detection of condensate proteomes is limited to a few mass spectrometry and proximity labeling studies: purified nucleoli[12,13], P-bodies[14] and stress granules[15,16] were subjected to mass spectrometry analysis for enrichment. Systematic experimental testing of protein members of condensates remains a bottleneck in the field. Therefore, computational methods can facilitate the process of characterizing proteins involved in biomolecular condensates at proteome-scale.

The main limitation of computational method development is related to the sparse experimental data of verified condensate-forming proteins. The first generation of computational models of phase separation were proposed based on properties of a few protein families, such as CatGranule[17] and PScore[18,19]. While these methods aided the discovery of novel condensate-forming proteins with similar properties, they do not generalize well[20].

In recent years, several data driven and machine learning based liquid-liquid phase separation (LLPS) predictors have been developed[21–24] making use of the experimental data aggregated in four LLPS databases[25–28]. While most predictors focused on identifying self-phase separating proteins (i.e. form in vitro condensates, that are also often drivers)[22–24] and thus trained on in vitro data, a recent metapredictor combined scores of previous methods and microscopy data to identify all condensate members (both drivers and clients)[29]. These methods have excellent performance compared to the first generation of predictors, nevertheless they have several shortcomings. For example a sub-optimal or biased definition of the negative datasets based on a priori assumptions about driving features, such as disorder being the main determinant, and accordingly using structured proteins from PDB as negative data[23]. Since no gold-standard negative dataset exists (i.e. proteins that do not form condensates), it remains a challenge for supervised methods to define an unbiased training dataset.

Here, we present a machine learning model called PICNIC (Proteins Involved in CoNdensates In Cells) to identify proteins involved in biomolecular condensates. As amino acid composition bias and patterning of charges were shown to impact the ability of proteins to form condensates[30–32], we design features that represent short and long range co-occurrences of amino acids in the protein sequence and structure (AlphaFold2 models), while including sequence complexity[23] and disorder scores[33] that were previously shown to be successful in identifying drivers. For training, we use a curated and non-redundant dataset of in vivo condensates (derived from CD-CODE)[34], and define the non-condensate forming, negative dataset, based on a protein-protein interaction network. Experimental validation of 24 proteins spanning a wide-range of structural disorder and biological functions suggests an ~88% success rate in identifying condensate forming proteins. Proteome-wide predictions by PICNIC estimate that ~40–60% of proteins partition into condensates across different organisms, from bacteria to humans, with no apparent correlation with organismal complexity or disordered protein content. Precomputed scores on 14 species are available at picnic.cd-code.org and the picnic-bio Python package can be used to compute scores for any sequence of interest.

## Results

### Defining condensate-forming proteins

In order to develop a model to identify condensate-forming proteins, we assembled a ground truth dataset for *H. sapiens*, that has the most experimentally studied condensates of all organisms, to date. Since we aimed at developing a binary classifier, we considered two classes of proteins: (1) proteins involved in condensates (positive dataset) and (2) proteins not involved in condensates (negative dataset) (Fig. 1a). The positive dataset was constructed from a semi-manually curated dataset of biomolecular condensates and their respective proteins, called CD-CODE (CrowDsourcing COndensate Database and Encyclopedia),

developed by our labs[34]. CD-CODE compiles information from primary literature and from four widely used databases of LLPS proteins[25–28].

Building the negative dataset is a complicated task as there is no publicly available resource that reports proteins that do not form condensates. Additionally, condensates may form only under specific conditions[35]. Here, we defined the negative dataset based on protein-protein interaction network (InWeb database[36] for human proteins). We excluded all proteins that have direct connections with known condensate proteins. We reasoned that these proteins are potential condensate members that have not yet been studied. The remaining proteins comprised the negative dataset (Fig. 1a). Of course, this procedure doesn't guarantee the absence of condensate proteins among the negative dataset (false negatives). But exclusion of proteins that directly interact with proteins that were reported as members of synthetic or biomolecular condensates is lowering the probability of mixing positive and negative data. Overall, our non-redundant dataset (filtered by 50% sequence identity) contained 2142 positive and 1709 negative human proteins, which were divided by 4:1 ratio into training and test datasets.

### PICNIC identifies sequence- and structure-determinants of condensate formation

We hypothesized that the ability to form condensates is encoded in the proteins' sequence and structure, and developed a machine learning classifier called PICNIC (Proteins Involved in CoNdensates In Cell) based on sequence-distance and structure-based features derived from AlphaFold2 models (Fig. 1b), in total 65 sequence-distance-based and 21 structure-based features.

It was shown that many proteins involved in condensates harbor intrinsically disordered regions (IDRs) and low-complexity sequences. IDRs, due to their inherent flexibility, multi-valency and ability to sample multiple conformations, are adept at a wide array of binding-related functions including molecular assemblies[9,37,38]. Therefore, we also tested several metrics of disorder and sequence complexity as features (see Methods). Our final model contained several features related to disorder, such as IUPred scores[33], that have a feature importance of 0.5–3%.

Although the presence of highly disordered residues is among the most important features (Fig. 1d, pink), it is not a prerequisite for a protein to have long disordered domains to be a member of a condensate(s). This is supported by the observation, that the proportion of known condensate-forming proteins with no disordered regions in the human proteome is 21% (disordered regions <10aa. Fig. S1), while 33% of all human proteins have no disordered regions. For example, Human protein Guanine nucleotide exchange factor C9orf72 is a driver protein in stress granules; Speckle-type POZ protein is a driver in nuclear speckle and SPOP/DAXX body. Both proteins consist of ordered domains that were experimentally determined by electron microscopy and X-ray crystallography, respectively (Fig. S1, PDB ids 6LT0 ad 3HU6). Thus, both the analysis of experimentally verified condensates and the selected features by our model suggest that disorder is not a necessity for condensate-forming proteins.

Along with overall sequence complexity and disorder scores of a protein, the secondary structure of individual residue types was also found to be important. We used the confidence score of the AlphaFold2 model prediction, the pLDDT score, that was shown to correlate with sequence disorder[39]. We represented the occurrence of an amino acid (AA) in a given secondary structure element (SSE) with a given model confidence as a triad (AA-SSE-pLDDT).

As amino acid composition bias and patterning of charges were shown to impact the ability of proteins to form condensates[30–32], we developed features that represent short and long range co-occurrences of amino acids in the protein sequence. We represent co-occurrence of amino-acids in the protein sequence within a distance (number of amino acids in linear sequence) by triads ($AA_L$,

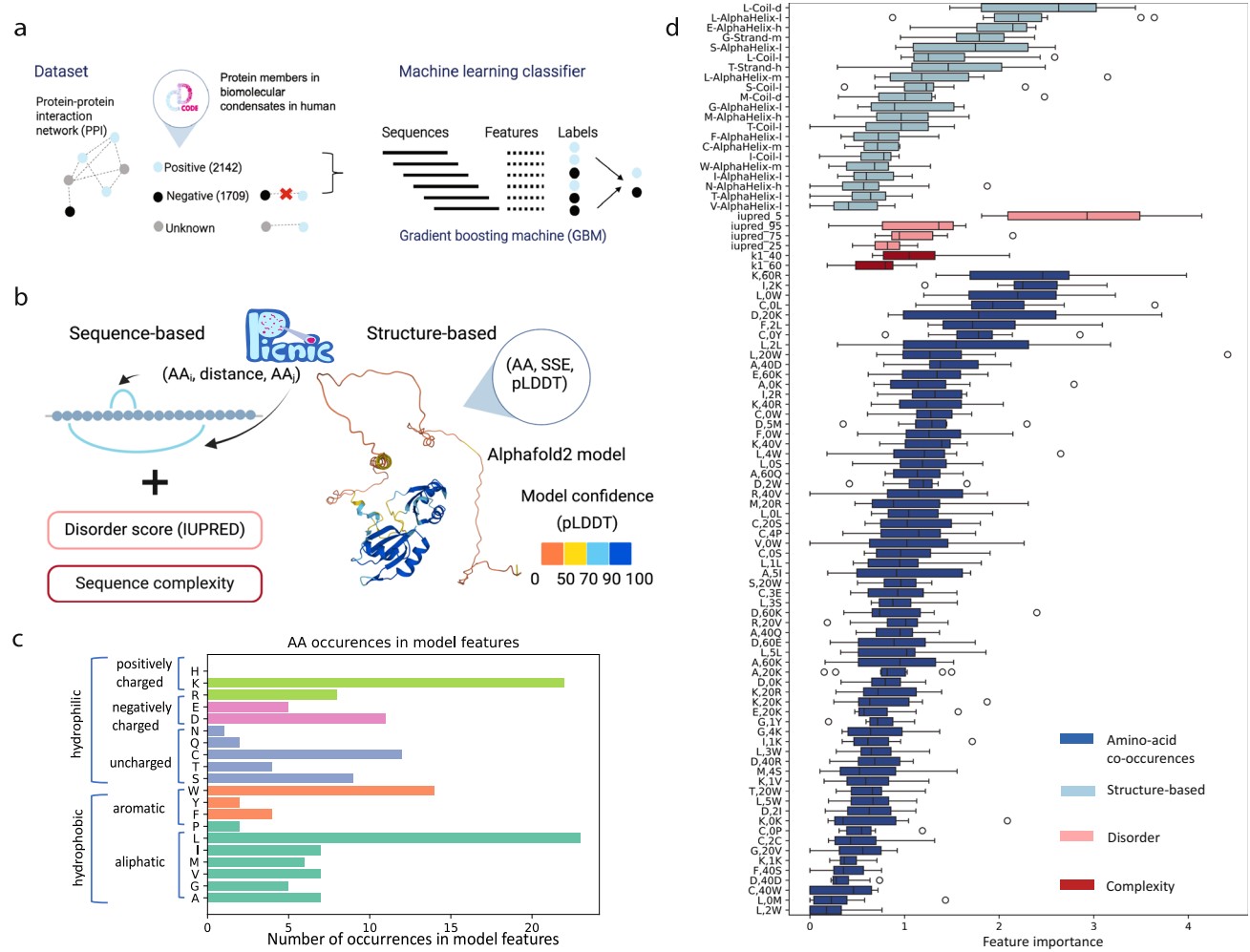

**Fig. 1 | Development of PICNIC (Proteins Involved in CoNdensates In Cells) algorithm. a** In order to construct a training dataset, we annotated the known condensate-forming proteins from CD-CODE[34] (positive dataset, members of bio-molecular condensates) on the protein-protein interaction (PPI) network, and we excluded their first connections (proteins having interactions with condensate proteins). The remaining proteins comprised the negative dataset. Gradient boosting machine was used to distinguish two classes of proteins: members of biomolecular condensates and proteins that are not involved in any type of bio-molecular condensate. **b** Sequence, structure and function-based features of PIC-NIC. Sequence-based features included sequence complexity, disorder score (IUPred), and features based on amino acid co-occurrences. Structure-based fea-tures based on AlphaFold2 models included the pLDDT score, a per-residue mea-sure of local confidence on a scale from 0 to 100 (colored on the structure). We annotated the secondary structure (SSE) based on 3D protein structures using STRIDE and all possible triads in the form (AA, SSE, pLDDT) were calculated. **c** Amino acid occurrences in the features of PICNIC model show that Leucine and Lysine contribute most to the model predictions. **d** Feature importance of PICNIC is consistent across different folds ($N = 10$). The boxes show the quartiles of the dataset, where first black horizontal line of the rectangle shape is first quartile or 25% the second black horizontal line is the second quartile or median, the third black horizontal line is third quartile or 75%. The whiskers extend to points that lie within 1.5 IQRs (interquartile range) of the lower and upper quartile, the outliers are displayed as circles. Features constitute four groups: based on AlphaFold2structures (light blue), disorder (pink), complexity (dark red) and amino acid co-occurences (blue). Source data are provided as a Source Data file.

*distance, AA₂*). After feature selection, the long-range distance between charged amino acids, e.g. Lysine and Arginine (K,60, R) and Aspartic acid and Lysine (D,20,K), and short-range distance of Leucine and hydrophobic amino acids (e.g. L,0,W; F,2,L; L,2,L), and the distance between Cysteine and hydrophobic amino acids were shown to be the most important features (Fig. 1d). Among the amino acids, Lysine and Leucine amino acids contribute the most to the model (Fig. 1c).

**PICNIC accurately identifies proteins involved in biomolecular condensate formation**

Several data-driven predictors were developed in the last few years, that aim to predict proteins involved in LLPS from protein sequence alone or from sequence and experimental data, such as microscopy

images[40]. Here, we compared the performance of PICNIC to sequence based predictors, PSAP[23], DeePhase[41] and the general model of PhaSePred[29] (PdPS-8fea based on 8 features) (Fig. 2).

We compared the performance of tools on three different datasets: (1) test dataset from the recently published PhaSePred methods[29]; (2) proteins forming nuclear puncta defined by the OpenCell project[42]; (3) test dataset generated from CD-CODE[34] (see Methods, Dataset S1). Although the CD-CODE test data is not inde-pendent and was partially used by existing predictors during their training process, PICNIC has superior performance with a maximum F1-score of 0.81 (Fig. 2c).

To further validate our model, we used microscopy images from Human Protein Atlas (HPA) where fluorescently labeled proteins were

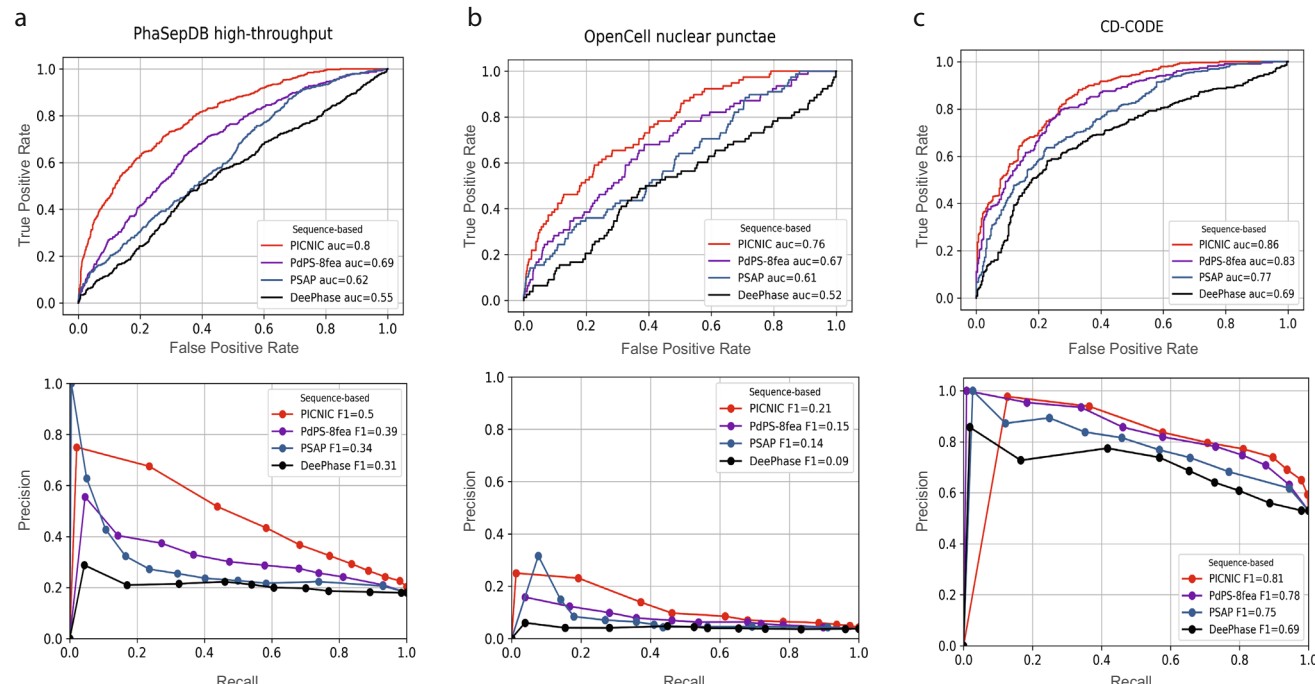

**Fig. 2 | PICNIC has the best performance in predicting condensate-forming proteins.** Comparison of sequence-based predictors (PICNIC, PdPS-8fea, PSAP, and DeePhase) of condensate proteins using different metrics. **a** Test dataset from PhaSepDB high-throughput retrieved from[29] (441 positive and 1998 negative examples, excluding proteins that were part of the PICNIC training set), (**b**) test dataset from OpenCell[42] (78 positive and 1998 negative examples excluding proteins that were part of the PICNIC training set), (**c**) test dataset from the current study based on CD-CODE[34] (338 positive and 299 negative examples, i.e proteins that were not part of the PICNIC training set). PICNIC outperforms sequence-based predictors even on the test set that includes training data of previously published predictors, that may inflate their performance. Source data are provided as a Source Data file.

imaged and their cellular localization was determined[43]. Specifically, three types of cellular localizations were screened: nucleolus, centrosome and nuclear speckle. We filtered the list of proteins from HPA that were already in our training set that resulted in 484 proteins with known localization. Overall, PICNIC scores were higher for the proteins from HPA than for proteins without known localization (Fig. S4). 69% of proteins mapped from HPA (with exclusion of the proteins from the training dataset) have a PICNIC score greater than 0.5, meaning that PICNIC correctly identified them as members of biomolecular condensates. It should be noted that HPA doesn't report if a protein does not belong to given condensate (negative examples). Therefore, this dataset can be used only to check model sensitivity (recall, what fraction of true condensate forming proteins were predicted correctly), but not model precision (what fraction of positive predictions are actually true positives).

## PICNIC is robust in identifying small sequence perturbations that impact condensate formation

A challenging task for a computational predictor is to be sensitive to small sequence perturbations that can impact condensate formation. To test if PICNIC can distinguish similar sequences with altered condensate forming properties, we considered the synuclein family, that comprises three paralogs in human. Although they have similar sequences (60–70% identity, Fig. 3a, c) and structures as predicted by AlphaFold2 (Fig. 3b), only α- and γ-synuclein form condensates in vivo, and only α-synuclein phase separates in vitro. Specifically, FITC-labeled β-synuclein, which lacks the characteristic NAC region of α-synuclein, does not phase separate at high concentrations (200 μM) and under crowding conditions (10% [*weight/volume*] PEG), whereas FITC-labeled α-synuclein forms condensates under the same conditions[22,44,45]. While α- and γ-synuclein can form amyloid-like fibers, β-synuclein does not[45,46]. Moreover, α- and γ-synuclein are part of biomolecular condensates: α-synuclein is reported to be the member

of synaptic vesicle pool condensate[46], γ-synuclein is a member of the centrosome[47], but β-synuclein has not been found in any biomolecular condensates yet. PICNIC is the only method tested here which accurately predicts the in vivo condensate-forming ability of the synuclein family (Fig. 3d). Other methods either give the same score for all three paralogs and/or do not predict the correct tendency of condensate formation in vivo. The features that stand out in β-synuclein and are absent in the most important features of α- and γ− synuclein (I-Alpha helix-l, F-Alpha helix-l) are connected to hydrophobic amino acids, being part of alpha-helix with low pLDDT score (Fig. S16). Thus, the structural changes involving the alpha helix are likely to drive the signal. We surmise that PICNIC is sensitive to structural rearrangements of proteins, and hypothesize that the bending of alpha-helix in β-synuclein potentially hinders the protein's ability to form condensate as highlighted on the structure (Fig. S16).

This encouraged us to further test, if PICNIC can predict the impact of mutations, i.e. substitutions and deletions. We assembled a dataset of sequence perturbations that impact the ability of a protein to form condensates described in the literature. We excluded the most commonly used phase-separating proteins such as FUS because they are used as a model for many algorithms, and the performance of these proteins is heavily biased. To this end, our dataset comprised of proteins with 27 single mutations and deletions in 10 proteins in total (Table S1, Dataset S5). PICNIC scores were consistently lower for mutant sequences that have a reduced or completely abolished ability to form condensates (Fig. 3e, f). However, the scores are still higher than the threshold 0.5. This indicates that the impact of mutations is better resolved when interpreted in relation to each other, since the overall ability of the protein to form condensates is encoded in features that are shared across the different mutants. Nevertheless, PICNIC can predict the relative ability of mutants to form condensates.

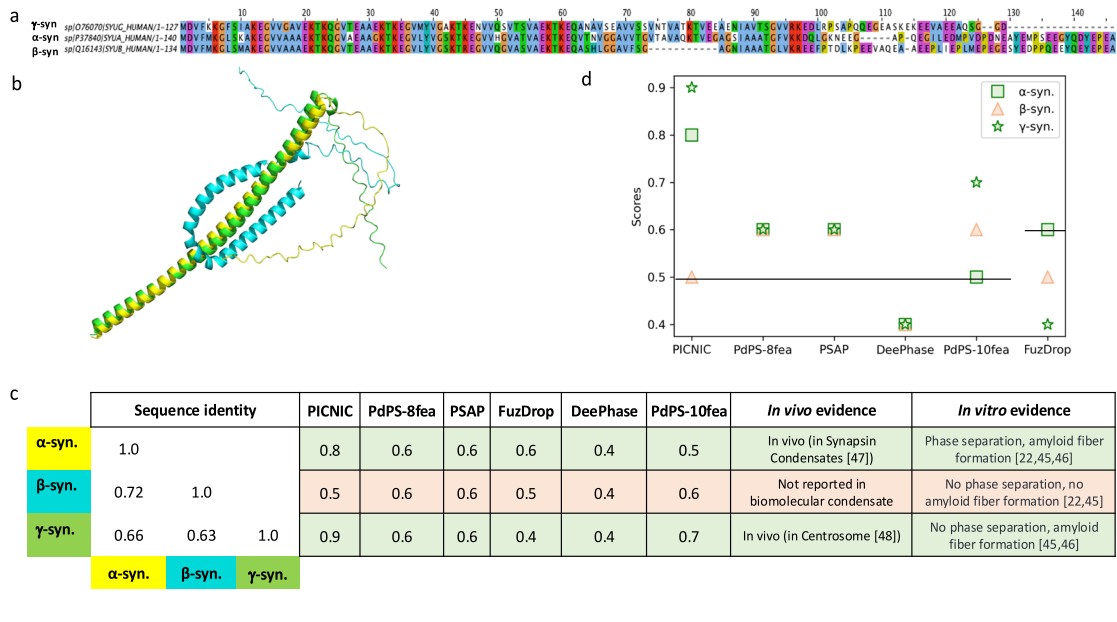

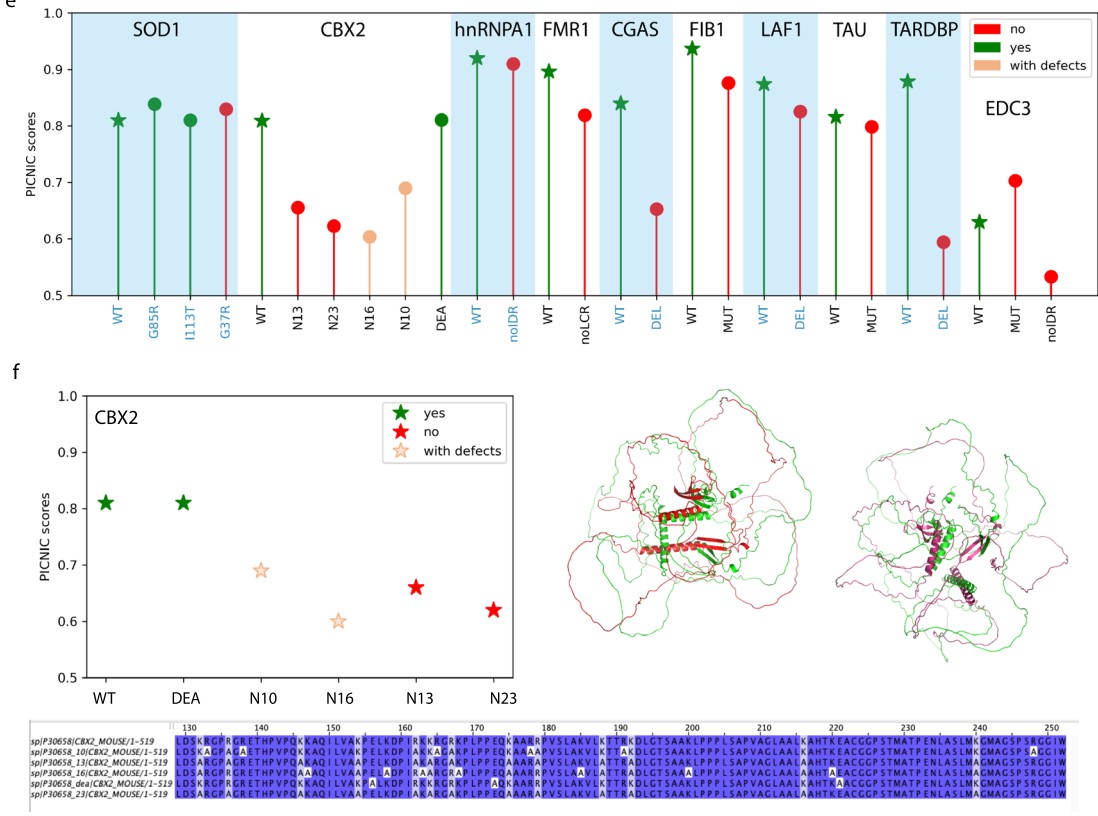

**Fig. 3 | PICNIC captures the different condensation behavior of paralogs and mutant sequences. a** The three paralogs of the synuclein family in human share high sequence identity as depicted in the multiple sequence alignment. **b** Structural models for α-synuclein (yellow), β-synuclein (cyan) and γ-synuclein (green), predicted by AlphaFold2 reveal that β-synuclein has a bent structure. **c** Despite the high sequence similarity, only α- and γ-synuclein are part of biomolecular condensates, while β-synuclein has not been found in any biomolecular condensates yet and was shown not to phase separate in vitro. **d** Comparison of prediction scores of different tools in identifying condensate forming (α and γ, green) and non-condensate forming paralog (β, red). PICNIC accurately predicts the condensate-forming ability of the synuclein family, and ranks β-synuclein the lowest, while other tools give equivalent scores to all paralogs or fail to identify the right trend. Vertical lines indicate the threshold used by the various methods to

classify condensate-forming proteins. **e** PICNIC scores of WT (shown as stars) and mutant sequences assembled from the literature (Table S1). **f** Example of PICNIC's performance on the mutated sequences of CBX2[73]. Whereas the scores for the canonical sequence and mutant CBX2_DEA, that both form condensates (green stars) are high, the score decreases for the mutants with reduced ability to condensate (empty red stars) (CBX2_N10) and (CBX2_N16), and for the mutants CBX2_N13 and CBX2_N23 that do not form condensates (red stars). Sequence alignment of the canonical sequence of CBX2 and the mutated sequences studied here. On the left panel the structural alignment between CBX2 (green) and CBX2_N13 (red), as well as CBX2 (green) and CBX2_N23 (purple) points out that even with preserved SSE, their 3D orientation affects the proteins' property to condensate. Source data are provided as a Source Data file.

## Experimental validation of predicted condensate-forming proteins

In order to experimentally validate our model, we decided to predict the condensate localization of poorly characterized human proteins and sought to validate their condensate-forming behavior inside living cells. To do so, we chose 24 proteins which: (1) cover diverse molecular functions spanning the entire central dogma of molecular biology and regulation of all major cellular bio-polymers for instance, nucleic acids, proteins and chromatin (Fig. S8), (2) represent the average sequence length of human proteins (i.e., around 350 amino acids) by having a range of 125 to 684 amino acids, (3) represent diverse 3D structures from ordered, alpha-helical, beta-stranded to highly disordered (Fig. 4b) and (4) are known to be involved in genetic diseases (AIMP1, CWC27, RP9, LMOD1) as well as host-pathogen interaction (IF2GL). Overall, the 24 proteins (Dataset S2), we chose for experimentally verifying and benchmarking PICNIC, represent global cellular functions and therefore are suitable to demonstrate how robust our machine learning model is in predicting condensate-forming proteins across entire proteomes.

We cloned 24 transgenes and transfected them in U2OS cells expressing fluorescently labeled proteins (see Methods). Using fluorescent imaging, we found, that 21 out of the 24 tested proteins (87.5%) localized to mesoscale foci without any stressors, while 3 proteins (C1ORF52, SPAG7 and CWC27, encircled in red) localized to the nucleoplasm without forming any discernible mesoscale foci (Fig. 4a, Fig. S10). Foci were defined based on enrichment in fluorescent intensity, i.e., the intensity ratio inside relative to outside the foci is greater than one (Fig. S9a). In sum, only 3 tested proteins tested show close to no detectable foci, and 21 form foci (Fig. 4a).

In order to classify the observed foci as biomolecular condensates, we aimed to define quantitative characteristics and thresholds. We measured four simple characteristics from fluorescent microscopy images (Fig. S9): area and perimeter, informing on the size and the typical number of proteins in a foci; shape (roundness); number of foci per cell. Next, we decided on a threshold for these characteristics to aid a quantitative definition of condensates. We consider foci as condensates above the diameter of 350 nm (distance between two furthest pixels in one condensate), that is well above the diffraction limit. This would correspond to at least ~1 μm perimeter assuming a near round shape (Fig. S9c). Using back-of-the-envelope calculations, we can consider an average protein size as 10 nm³, then a 1 μm³ compartment can contain ca. 1 million protein molecules and a 500 nm³ compartment can contain 100,000 protein molecules. We note, that other super resolution techniques are required to characterize the size of the clusters of proteins below the diffraction limit.

## Experiments confirm 87.5% of PICNIC predictions

By applying the above definition, in our dataset, 75 %, i.e., 18 proteins (encircled in blue) form high confidence condensates, 12.5 % i.e. 3 proteins (encircled in orange) form low confidence condensates (foci with perimeter <1 μm) while other 3 are not forming any condensates (encircled in red) and exhibit a fluorescent intensity ratio ~1. We observe most condensates to be round (Fig. S9d). The number of condensates per cell varies between 1 and 100 s depending on the protein of interest.

Next, we wanted to characterize the localization of the condensates that the proteins formed. We observed that 9 proteins (TYW5, SPA24, AIMP1, ZC3H15, IF2GL, LMOD1, RPS4Y2, DRC4, RS10L) localized to cytoplasmic bodies and 7 (RAD51AP1, KHDC4, CWC25, POLD3, RAMAC, DRC4, RP9) localize to nuclear bodies (Fig. 5a, Fig. S10).

Using co-localization experiments with known nucleolus (Fibrillarin) and processing body (P-body) (DCP1a) markers, we found that many proteins (H2A1H and RAD51AP1, H1T, MRPL1, RS10L, RPS4Y2, PolD3) can localize to the nucleolus at least in some cells (Fig. 5a, see also Fig. S10), a well-characterized liquid-like nuclear condensate (FRAP experiments in Fig. 5b), and PHP14 can localize to P-bodies, another well-characterized cytoplasmic condensate (Fig. 5a).

H2A1H (in purple) shows rather weak localization as a rim around the DFC (in green) showing sub-nucleolar localization specific to the outer Granular center (GC, see the cartoon representation of the nucleolar architecture). RAD15-associated protein 1 (RAD51AP1) shows multi-condensate localization that varies from exclusive nuclear bodies (Localization 3 and 4), to nuclear bodies abutting the nucleolus (Localization 3), and RAD51AP1 forming rim around DFC (GC, Localization 1) to sub-nucleolar localization (Localization 4) as well as complete but weak nucleolar localization (Localization 2) (Figs. 5a, S10) suggesting an interesting role for this protein's involvement on multiple nuclear-condensates possibly in a cell-cycle stage regulated manner. The localization patterns of the condensate forming proteins are consistent with the wide-range of molecular functions that these proteins perform (Fig. S8). We also observed more than one type of foci in case of many proteins for e.g. DRC4: nuclear and cytoplasmic; MRPL1: nuclear and cytoplasmic bodies; RS10L: cytoplasmic bodies, filaments as well as nucleolar localization (Fig. S10).

In addition, FRAP recovery profiles for a subset of the condensate forming proteins (RAD51AP1, KHDC4, CWC25, RAMAC, DRC4, RP9, RBMY1D, TYW5) revealed, that 6 out of 8 tested proteins show very fast dynamics indicated by the recovery of the bleached foci, while two (TYW5 and RBMY1D) shows little to no recovery at the indicated time (Fig. 6).

In sum, we do not see any correlation with disorder content of a protein and its ability to form condensates in our experimental dataset (Fig. 4c). Other popular tools to predict condensate proteins would fail to make correct predictions for many of these proteins as shown by the high misclassification rates (Fig. 4d). Overall, 87.5% of PICNIC predictions were found to be correct (misclassification rate is 25% for high confidence condensates and 12.5% if we include both high and low confidence condensates) in our experimental assays validating the model.

## Proteome-wide predictions detect no correlation of predicted condensate proteome size with disorder content and organismal complexity

To demonstrate the generalizability of PICNIC, which was trained on human data, we tested its performance in identifying known condensate-forming proteins of other organisms. We screened the CD-CODE database[34] to evaluate the fraction of proteins that were correctly identified as members of condensates by our predictor. PICNIC successfully predicted 72% of such proteins in mouse and 86% in *Caenorhabditis elegans* for example (Fig. 7a). Thus, PICNIC model is species-independent and is useful for different organisms to assess the ability of proteins to be involved in biomolecular condensates.

To estimate the overall proteome fraction of condensate-proteins, we calculated PICNIC scores for 26 different organisms across the tree of life including bacteria, plants and fungi (Fig. 7). We selected 14 organisms that have already known condensate protein members that were experimentally verified in CD-CODE (we excluded organisms from further analysis where the number of known proteins is too small to compute statistics on the performance: *Danio rerio* ($N = 14$), *Dictyostelium discoideum* ($N = 2$), *Escherichia coli* ($N = 6$), *Mycobacterium tuberculosis* ($N = 1$), *Oryza sativa* ($N = 1$), *Candida albicans* ($N = 6$)). PICNIC correctly identified 50–100% of the known condensate proteins across organisms in CD-CODE (Fig. 7a). Although PICNIC was trained on human data, it generalized well and is likely applicable to proteomes of other organisms.

Next, we performed proteome-wide assessment of condensate proteins across 26 organisms. We found that the proportion of the predicted condensate-forming proteome is 40–60%, and is similar across related organisms, e.g., 42% and 39% in human and mouse,

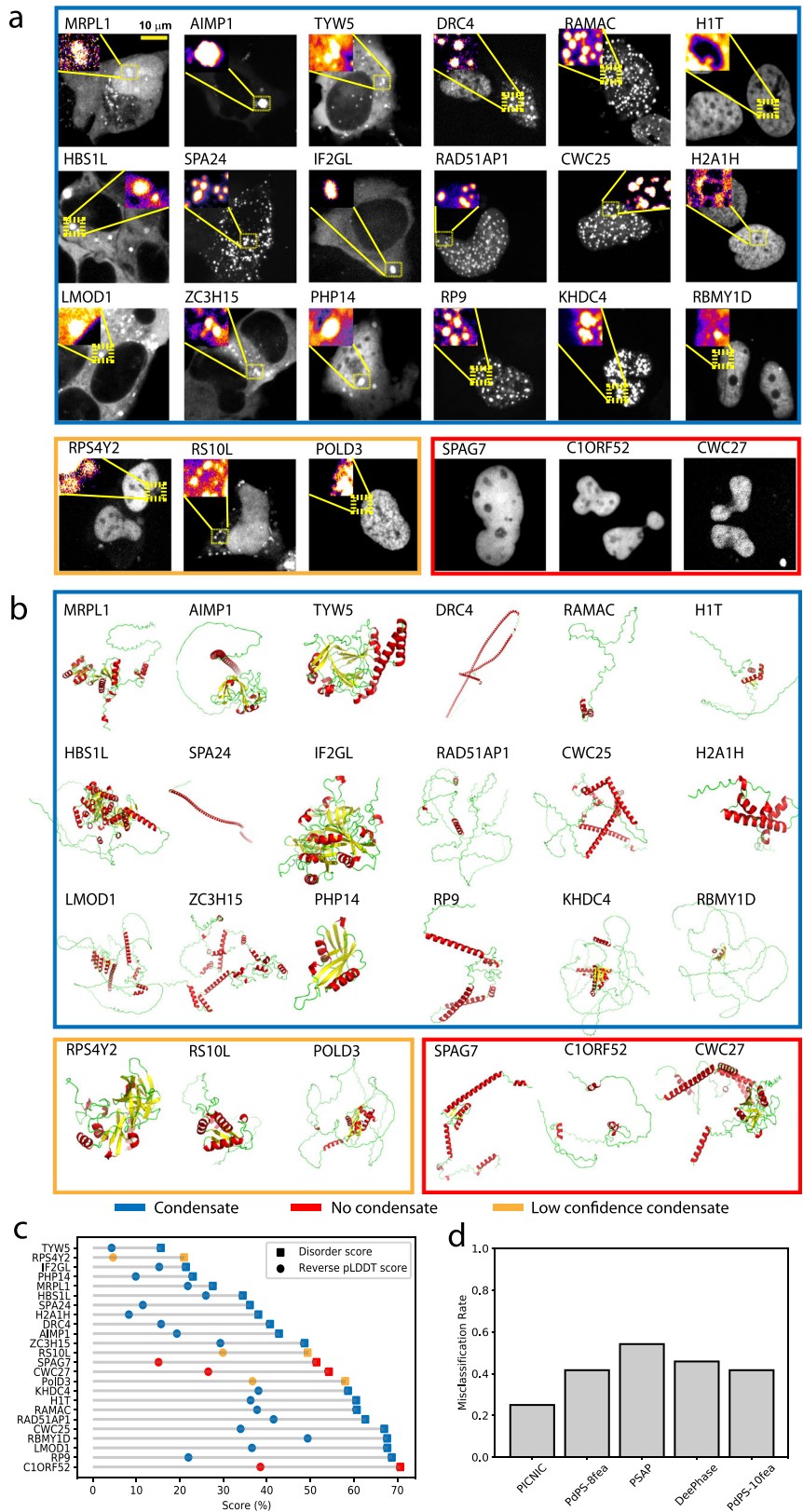

**c**
Disorder score ■
Reverse pLDDT score ●

**d**

Condensate ■   No condensate ■   Low confidence condensate ■

respectively (Fig. 7b). Interestingly, while the fraction of disordered proteins increases with organismal complexity as shown before[48,49], we found no correlation between fraction of predicted condensate proteins in a proteome and the disordered protein content (Fig. 7c) across the 26 species tested even when using different metrics to assess the disorder of a proteome (Fig. S14). For example, *E. coli* and *H. sapiens*

have both ~40% of their proteome predicted to be involved in biomolecular condensates (Fig. 7b).

## Discussion

How protein sequence encodes condensation behavior has been an open question. Here, we present a machine learning classifier that can

**Fig. 4 | Most (18 out of 24) tested proteins form high confidence condensates *in cellulo*. a** Representative images of the U2OS cells expressing the tested proteins, N-terminally tagged with an iRFP fluorescent tag. Formation of mesoscale cellular condensates are highlighted in the inset. All images are scaled to the scale bar 10 µm (shown on upper left image). We found 21 out of the 24 tested proteins (87.5%) formed mesoscale foci without any stressors, while 3 proteins (C1ORF52, SPAG7 and CWC27, encircled in red) localized to the nucleoplasm without forming any detectable foci (foci were defined by exhibiting a fluorescent intensity ratio >1). Notice the presence of rim-like structures in case of H1T and H2A1H. Using size, shape and the fraction of cells forming mesoscale foci as a deciding characteristic, ~75% i.e., 18 proteins (encircled in blue) form high confidence condensates, and 3 proteins (encircled in orange) form low confidence condensates (foci with longest diameter <350 nm. Fig. S9). The experiments were repeated at least twice to ascertain the reproducibility of the results. More representative images are shown in Fig. S10 and the raw images are provided as Dataset S4. **b** Wide range of secondary structural motifs covered in the test proteins; AlphaFold2 structural models of the proteins are colored according to secondary structures. Notice the wide range of structural motifs, alpha-helical (red), beta stranded (yellow) to largely disordered (green) proteins (AlphaFold2 structures are provided as Dataset S6). **c** Disorder content (computed as mean IUPred score or reverse pLDDT score (1 - pLDDT)) of the tested protein does not correlate with the ability to form condensates. **d** Comparison of the predictions provided by sequence-based predictors (PICNIC, PdPS-8fea, PdPS-10fea, PSAP, and DeePhase) of protein condensates. PICNIC exhibits the lowest misclassification rate for the tested 24 proteins. Source data are provided as a Source Data file.

learn and decode biomolecular condensate forming behavior of proteins, that goes beyond protein structural disorder. Our model, PICNIC was trained yet on the largest positive and an unbiased negative dataset for the prediction of proteins involved in condensates in vivo. It reaches precision of 77% (81% for recall) at the suggested score threshold of 0.5 (Fig. 2) outperforming previously developed methods. PICNIC may shed light on several sequence features important for condensates: (1) the importance of IDRs in condensate member proteins without being a necessary pre-requisite, (2) secondary structure of individual residue types and (3) amino acid composition bias and co-occurrence in the protein chain.

The success of PICNIC model relies on several innovations: (1) amino acid co-occurrence features combined with protein structure-based features, that became only possible on a proteome scale since the AlphaFold2 revolution[50,51]; (2) gradient boosting classifier that has an appropriate model complexity that fits the size of the training dataset; (3) improved curation and definition of the positive and negative datasets. Specifically, PICNIC models benefit from high quality positive data, that is the manually curated database of biomolecular condensates CD-CODE[34]. Additionally, we designed the negative dataset based on no a priori assumptions about protein disorder/structure. Previous predictors used either a (i) exclusion of positive dataset from all known proteins[29,41] or (ii) proteins with 3D structures (retrieved from PDB) since phase separation is common in disordered proteins that do not have well-defined 3D structures. The simple exclusion does not provide reliable negative dataset, as it may contain many potentially condensating proteins that have not been discovered yet. The second approach generates a biased negative dataset and is problematic because of two reasons: (1) condensating proteins can have well-defined structure (Fig. S1), (Fig. 4b, c), (2) such dataset is biased towards set of properties inherited by experimentally solved proteins. E.g., disorder predictors would also use 3D structures as negative dataset. To resolve this issue, we used a protein-protein interaction network-based approach and excluded proteins that have a connection in the network with known condensate proteins (Fig. 1a).

PICNIC is based on gradient boosting machines (GBM), which allow optimization of different of loss functions which provides necessary flexibility, but more prone to overfitting. Here, the latter was compensated by choice of tree depth and by providing an evaluation dataset. GBM generally outperforms simpler models such as Random Forest or Support Vector Machine, but at the same time it doesn't require as much data as models based on Neural Networks. Because there is not enough well-annotated data for positive and negative datasets, we chose GBM to mitigate the tradeoff between model complexity and its performance. Using a simple model allows investigating which features drive condensation. Extracting the most important features for each individual protein of the algorithm can give a hint about properties, specific for this particular protein, that are important for its condensation. For example, the features highlighted by PICNIC for FUS protein condensation (Fig. S15) are in agreement with a study which shows that phase separation is primarily governed by multivalent interactions among tyrosine residues from prion-like domains and arginine residues from RNA-binding domains, which are modulated by negatively charged residues[19]. Glycine residues enhance the fluidity, whereas glutamine and serine residues promote hardening. Thus, analyzing PICNIC features can generate hypothesis for potential mutations that may alter condensation.

We tested the ability of PICNIC to predict the impact of sequence perturbations on condensate formation. PICNIC correctly predicted that only α- and γ-synuclein can form condensates, and β-synuclein does not despite having high sequence identity to its paralogs (Fig. 3, Fig. S16). PICNIC scores were also consistently lower for mutant sequences that have reduced or no ability at all to form condensates based on a set of experiments assembled from the literature (Fig. 3e, Table S1). We want to note, that other prediction tools could not be applied to compute scores for mutated protein sequences. Specifically, the state-of-the-art meta-predictor Phasepred, although it provides great web-server for prediction, it accepts only gene name or Uniprot id as inputs, thus allowing computations for WT sequences. Other tools that we benchmarked against require Uniprot id, or are unable to run off-the-shelf for mutant sequences, thus we could not compare them to PICNIC. Nevertheless, the relative PICNIC scores to a WT sequence can be used to generate hypothesis about the impact mutations may have on a protein's ability to form condensates.

As a proof of concept, we explored the possible means of increasing the performance as well as resolution of our model by integrating additional functional information that is already known about each protein as described in Gene Ontology (GO) terms (Fig. S2). We developed a second classifier, called PICNIC_GO that combines GO terms and the previously used PICNIC features (Fig. 1b). After feature selection only 18 features were included in the final model, 10 new GO features and 8 features of PICNIC described in the previous section. We found, that the most significant GO term is RNA binding, which is superseding the importance of other terms and features by several orders of magnitude (Fig. S2b).

While the GO annotation feature is biased by existing knowledge, it nevertheless validates the design of the negative dataset: it highlights RNA binding molecular function as one of the of the most important features in PICNIC_GO model, which is known to play crucial role in biomolecular condensate formation, as RNA molecules are significant constituents of condensates[9,52]. This feature is efficient in discriminating the two classes of proteins because the positive and negative datasets show different distributions of RNA binding annotation. In PICNIC_GO the following functions were marked as the most important: transferase activity, transferring phosphorus containing groups, enzyme binding, phosphoric ester hydrolase activity, organic cyclic compound binding, and heterocyclic compound binding (Fig. S2b). The success of including functional annotation demonstrates that, for proteins with unknown cellular compartmentalization, specific set of functional descriptions can further improve the prediction their tendency to localize to the biomolecular condensates.

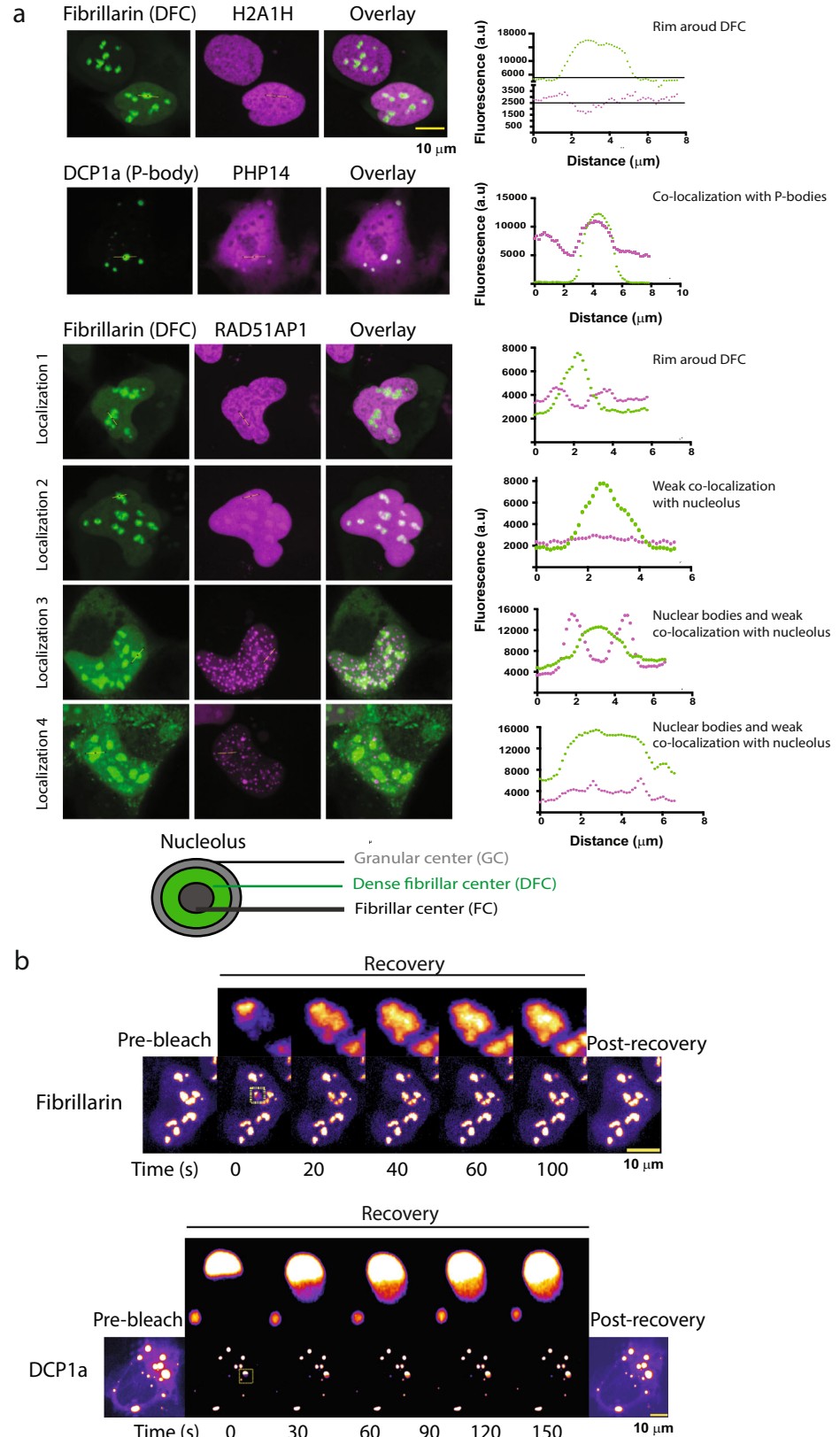

To explore the generalisability of our models, we compared the most important features learnt by PICNIC vs. PICNIC_{GO}. First, we calculated the most divergent gene ontology features between the dataset of known condensate-forming proteins (CD-CODE) and the dataset of whole proteomes of corresponding organisms (Fig. S5). Further, we compared these distributions to the distributions of potential condensate proteins predicted by PICNIC. This analysis demonstrated that PICNIC can recognize the protein properties deduced as most important by PICNIC_{GO} model, as well as terms describing cellular localization which were excluded from PICNIC_{GO} features (Fig. S6).

We also compared the two models in the opposite direction to see whether PICNIC_{GO} model can identify the sequence- and structure-

**Fig. 5 | A subset of the tested proteins localizes to known condensates. a** Co-localization of the cellular condensate-forming proteins with well-characterized liquid-like cellular condensates as can be concluded from the fluorescence intensity profiles correlation with the marker protein fluorescence profiles (shown in green). While RAD51AP1 (in purple) localizes strongly around the Dense fibrillar center (DFC, in green) forming a rim like structure, H2A1H (in purple) show rather weak localization as a rim around the DFC (in green) showing sub-nucleolar localization specific to the outer Granular center (GC). See the cartoon representation of the nucleolar architecture. Further, PHP14 (purple) co-localizes with the DCP1a- labeled (green) processing bodies. RAD15-associated protein 1 (RAD51AP1) shows localization that varies from exclusive nuclear body like appearance (Localization 3, 4) to RAD51AP1 forming a rim around DFC part of the nucleolus (GC, Localization 1),

abutting the nucleolus without forming a rim (Localization 3) and sub-nucleolar localization (Localization 4), as well as complete nucleolar localization suggesting an interesting role for this protein's involvement on multiple nuclear-condensates (see also Fig. S10) possibly in a cell-cycle stage regulated manner. All images are scaled to the scale bar 10 µm (upper right corner) and available as Dataset S4. Co-localization experiments were repeated at least twice to ascertain the reproducibility of the results (see also Fig. S10). **b** FRAP assays showing the fast recovery dynamics consistent with the liquid-like nature of the P-bodies (upper panel) and the Nucleolus (lower panel). Fibrillarin and DCP1a FRAP recovery profile, inset highlighting the fast recovery dynamics of targeted P-body and the nucleolus. The scale bar is 10 µm for each. Source data are provided as a Source Data file.

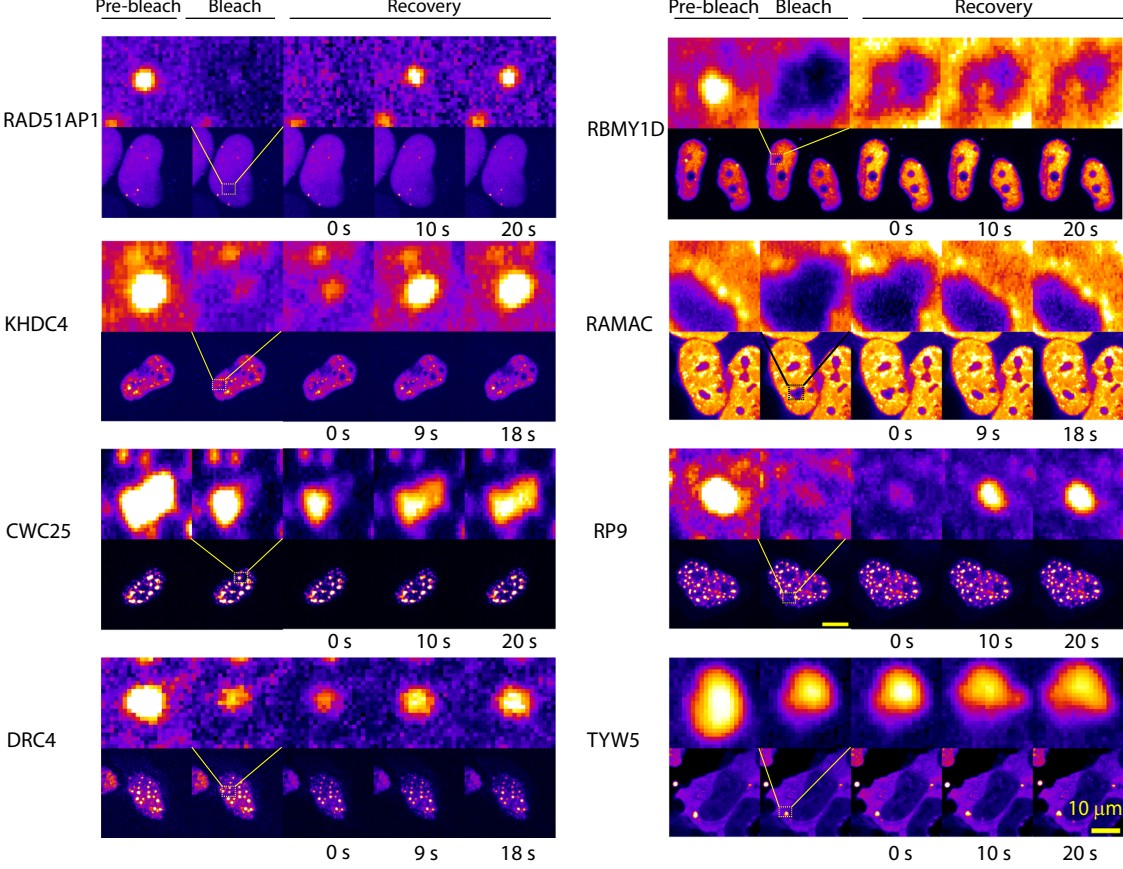

**Fig. 6 | FRAP assay suggests liquid-like condensates.** FRAP recovery profiles for the condensate forming proteins. 6 out of 8 tested proteins show very fast dynamics (at a time scale of 20 s, indicated) indicative of liquid-like nature, while two (RBMY1D and TYW5) show no recovery at the indicated time. All the images

with indicated whole nuclei (lower panel for each protein) are scaled to the 10 µm scale bar shown in the bottom right. FRAP experiments were repeated at least twice to ascertain the reproducibility of the results. Images are provided as Dataset S4.

based features considered as important by the PICNIC model. Interestingly, for different species different subsets of features were highlighted, but in concordance with features selected as most significant by PICNIC (Fig. S7). Thus, PICNIC that was trained without Gene Ontology annotation can detect properties of proteins in biomolecular condensates captured by Gene Ontology terms for different species and vice versa: PICNIC$_{GO}$ detects properties of proteins in biomolecular condensates captured by distance-based and AlphaFold-based features for different species, further validating the generalizability of the model. Overall, the propensity of a protein to be a member of biomolecular condensate is encoded in the sequence, and machine learning models, such as PICNIC can decode these properties.

Not surprisingly, the models that use existing experimental information outperform the sequence-based predictors (Fig. S3).

Specifically, PICNIC$_{GO}$ has the best performance (ROC-AUC = 0.91, F1-score = 0.84) followed by PdPS-10fea (ROC-AUC = 0.89, F1-score = 0.83). Nevertheless, both PICNICGO and PdPS-10fea are biased by the available information, GO annotation and microscopy data, respectively. Therefore, they are not applicable broadly, especially not for less well studies proteins and organisms.

While this paper was under review, a phase separation predictor, called PSPire[53] was published. We cannot compare performance on the human dataset, since PSPire was trained on all human data thus there is no independent benchmarking dataset that we could use. Nevertheless, we could compare performance on other species than human and found that PICNIC clearly outperforms PSPire and Phasepred on the tested mouse, yeast and *Arabidopsis* condensates (Fig. S13). Note, that the other tools used here for benchmarking do not provide scores

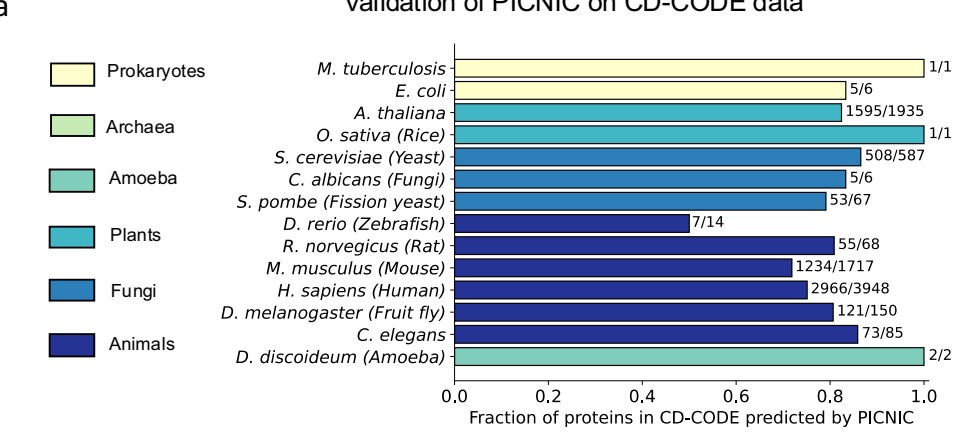

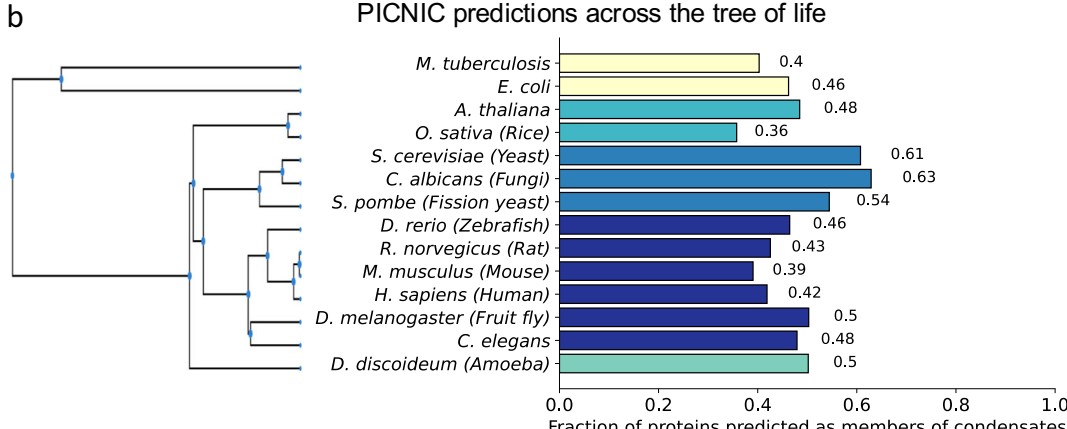

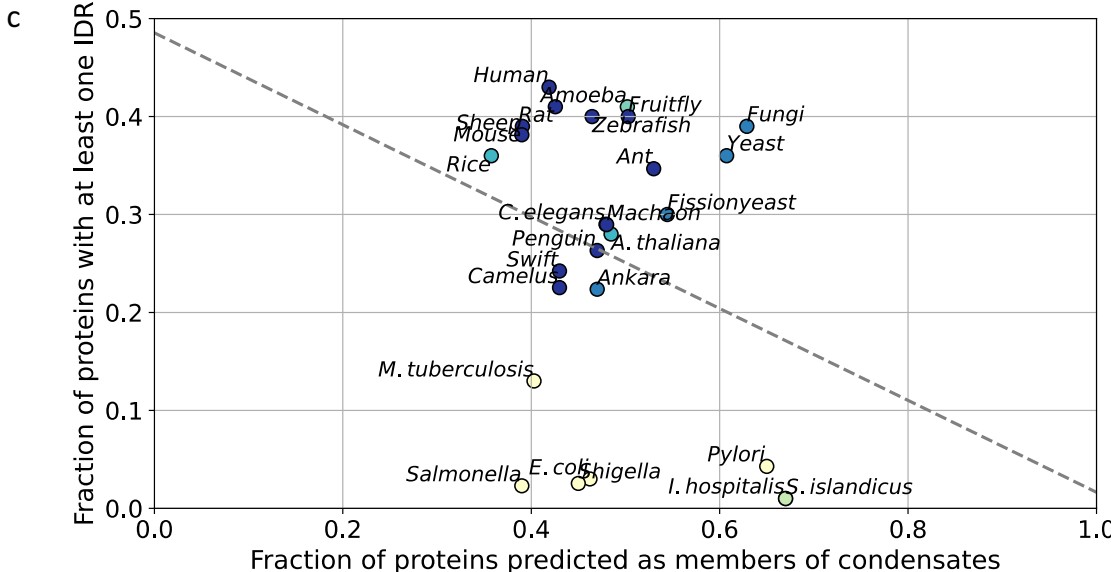

**Fig. 7 | Inferring condensate proteins across the tree of life reveals no correlation with disorder content. a** PICNIC model is species-independent. We validated the PICNIC model on known condensate proteins from 14 different species (defined by CD-CODE[34]). PICNIC correctly identified 70–100% of known condensate proteins of all species tested, except for zebrafish (50%). **b** Proteome-wide prediction of proteins in biomolecular condensates by PICNIC predictor. **c** Disorder content and fraction of condensate-forming proteins of a proteome are not correlated. The fraction of disordered proteins (proteins with at least one disordered region of > = 40 residues) in proteomes shows no correlation across 26 selected organisms from bacteria, archaea to mammals (Pearson $R^2$ = 0.08). Source data are provided as a Source Data file.

or do not offer computation of scores on new sequences, this is why we cannot report their performance. While PICNIC can be applied to any natural or synthetic sequence and is shared as an easy-to-use Python package.

We would like to emphasize that some of the other tools we benchmarked against were designed to solve a different task, namely to predict the ability of a protein to undergo phase separation or drive phase separation and in vitro condensate formation[22,41]. Specifically, PSAP[23] was trained on 90 human proteins that can drive phase separation, and FuzDrop[22] was parametrized based on 67 proteins known to self-phase separate based on in vitro experiments. Therefore, it is not expected that these tools would identify all condensate forming proteins, e.g. clients observed in vivo such as the nuclear punctae from the OpenCell project (Fig. S2b). In contrast, PICNIC was adapted to recognize the proteins that are present in biomolecular condensates (serving a role of a driver or client) regardless of the mechanism of condensate formation. Therefore, our model does not evaluate if a protein is part of synthetic condensates, i.e., in vitro experiments, but rather focuses on if a protein is part of condensates in biologically relevant conditions.

Interestingly, PICNIC scores for clients and drivers show similar distribution (Fig. S17). Since the same protein can behave as a driver or a client depending on the condensate identity and environment, we surmise that the ability of driving condensate formation is not solely encoded in the sequence and structure of individual proteins. This suggests, that additional input data are needed to be taken into account by the model. Studying the protein in the context of its interacting partners within a condensate may shed light on the properties of driver proteins.

While PICNIC is superior in predicting protein and its mutant's membership in biomolecular condensates, it has limitations. For e.g. it cannot directly predict which sequence regions are responsible for the condensate function in a given protein. FuzDrop for example, can recognize specific motifs in protein sequences that promote phase separation behavior[22,54–56]. However, the ability to predict the difference in the behavior upon mutations makes it a testable hypothesis generating platform to experimentally validate the regions in higher resolution, that are involved in the condensation.

The algorithms in this paper are designed to understand whether a protein has the potential to localize to a condensate, trained on previous experimental data. Here, we tested 24 predicted proteins and found that 87.5% form foci in cellulo, and 75% form high confidence condensates based on a quantitative definition of condensates. In making this analysis, we set a cut-off above the diffraction limit (Fig. S9). This does not mean that clusters of proteins under this limit do not form condensates, but that super-resolution techniques would be required to analyze this. Three proteins formed no condensates, nevertheless we cannot be certain that they would not form condensates under different conditions or cell-types.

Further, the number of cells with condensates can be highly variable. Such noise is common in biology, as often phenotypes are not fully penetrant. Even isogenic populations of cells and organisms show phenotypic variability, e.g. due to transcriptional noise[57]. We see the same phenomena with condensate formation, and therefore introduced the stringent criteria that high-confidence condensate forming proteins have foci in *most* imaged cells, while low-confidence condensates form foci in *few* cells only.

We showed that several of the tested proteins localize into known condensates, P-bodies or to the nucleolus (Fig. 5). It is important to state that the condensate localization makes no claim to biological function in cells. These condensates were observed in cell lines, using iRFP-tagged proteins that had varied expression levels. It is possible that some proteins will only form condensates under stress or in certain cell types at certain concentrations. In some cases, a protein for instance might not localize to a condensate under physiological

conditions, but might when overexpressed in cancer. Therefore, although each protein can localize to a condensate, it remains to be sorted out by detailed experiments whether any individual protein localizes to a condensate in a specific cell type at a certain concentration and using different tags and/or antibodies.

To further validate the ability of PICNIC to discriminate proteins that form condensates, we randomly selected 21 proteins predicted to be negative by PICNIC after filtering out the ones that are associated with membranes since these are "obvious negatives". We expressed the fluorescently tagged proteins in U2OS cells to visualize them and noticed, that 5 of them do not express at all (despite repeated attempts) and one localized to the ER. The remaining 15 proteins also showed weak expression. Overall, out of these 15 proteins, 10 formed no condensates, 2 formed low confidence condensates and 3 formed condensate-like foci (Fig. S11). Curiously, the protein previously reported not to form condensates in a recent paper[53], also formed low confidence condensates in our experiments (GTF2A2, Fig. S12). Previously the protein was expressed in HeLa cells tagged with GFP while in our experiments it was expressed in U2OS cells and tagged with iRFP. Thus, condensate formation of GTF2A2 can be cell-type or tag-dependent as it was shown to impact condensation across proteins[58], or even the concentration can be different in the two sets of experiments. In conclusion, PICNIC correctly identifies proteins that are less likely to form condensates, but cannot predict if they *never* form condensates.

In general, proving that a protein never forms a condensate is philosophically impossible, since this can be cell-type and condition-specific. This conundrum hinders current machine learning based efforts to train classifiers. We argue against a dichotomic view of condensate formation, in favor of a focus towards how likely condensation occurs under specific conditions. We assume that the proteins that are predicted to form condensates by PICNIC would robustly form condensates across cells (e.g. high confidence condensates). On the other hand, the proteins that are predicted *not* to form condensates should not form condensates in *most* conditions. In order to escape the puzzle of unobtainable negative data, we anticipate that future generations of condensate predictors will not just limit themselves to simple yes/no classification of condensate proteins, but will also incorporate condition-specific condensation.

We detect that structural disorder is not a prerequisite for condensate member proteins, as many of them have no IDRs both based on analysis of the CD-CODE database (Fig. S1b) as well as 5 out of the 21 proteins identified here experimentally as condensate members have <30% disordered residues (Fig. 4c). Thus, a wide-range of structural disorder can lead to condensate partitioning. Accordingly, we found no correlation of condensate proteome size and disorder content of an organism.

The generalizability of our model shows that the predictor learned general features (based solely on sequence information and structures that were also deduced from the sequence) across the tree of life. The provided results can shed light on evolution of biomolecular condensates across different species and by predicting condensate members beyond drivers can help identifying potential protein targets to modulate diseased biomolecular condensate behavior to aid drug design[59,60]. Overall, PICNIC accurately predicts proteins involved in biomolecular condensates and provides proteome-wide perspective on proteins involved in condensate formation in different species.

## Methods
### Construction of positive and negative datasets
The positive dataset of condensate forming proteins was extracted from CD-CODE database v1.00[34]. We used all human proteins with at least 1 evidence star as a positive dataset. The negative dataset was constructed by excluding proteins that interact with known

condensate forming proteins based on the InWeb v3 database[36]. After filtering the sequence for 50% sequence identity, our dataset composed of 2142 positive and 1709 negative proteins, that were divided by ratio 4:1 into training and test datasets. We divided the dataset into 20% test dataset (used only for testing the final model) and 80% working dataset, which was randomly divided into 70% training and 30% validation datasets for the 10-fold cross-validation (Dataset S1).

### Construction of dataset of proteins with mutations affecting their condensation ability

To date, there is no comprehensive resource of data reporting the mutations, that disrupt or induce the protein ability to condensate. The papers describing such cases are sporadic and are often reported for specific proteins only. There is no systematic study or database on how mutation will affect the protein ability to condensate, that would be generalizable to the whole class of the proteins involved in biomolecular condensates. Here, we gathered such cases, described in the literature, and excluded the most commonly used phase-separating protein FUS (because it is used as a model for many algorithms, the performance of these proteins is heavily biased). Our dataset comprised of proteins with single mutations[61,62] and deletions[28], 27 protein mutants for 10 genes in total (Table S1, Dataset S5). For each mutated protein, the Alphafold2 algorithm was used to model the structure of the protein, which was used as one of the inputs for PICNIC model.

### Model features

**Disorder score and sequence complexity.** Intrinsically disordered regions (IDRs) and low-complexity regions of proteins were shown to be an important feature for predicting the ability to phase separate. To estimate protein disorder, we used the IUPred algorithm[33], which assigns a score to each residue in the sequence. We used $k^{th}$ percentile with $k$ equal to [5,25,50,75,95], which is the score below which $k$ percentage of residue scores fall. The 5th and 95th percentiles were chosen instead of minimum/maximum values to exclude the bias due to outliers.

We calculated sequence complexity in order to identify low-complexity regions (LCRs). These regions often contain repeats of single amino acids or short amino acid motifs. We calculated sequence complexity according to the definition suggested by Wootton and Federhen[63] using two different sizes of sliding window for the protein sequence: 40 and 60. Proteins with length less than 60 residues were excluded from this analysis.

**Sequence distance-based features.** We represented the co-occurrence of amino-acids (*AA*) in the protein sequence within certain distance by a pair of triads ($AA_1$, $distance_{short}$, $AA_2$) and ($AA_1$, $distance_{long}$, $AA_2$), where $AA_1$, $AA_2$ are one of the 20 amino acids types, $distance_{short} \ni$ [1,2,3,4,5] and $distance_{long} \ni$ [0,20],([20,40)],[(40,60)],[60,80)]. These features represent short and long range co-occurences of amino acids in the protein sequence. Short threshold $distance_{short}$ defines the distance between two types of amino acids (e.g., 1 means neighboring residues). Long threshold, $distance_{long}$ defines a window equal to 20, that is the distance in sequence between the two types of amino acids co-occur. The total number of tested sequence distance-based features was 1890, including both short and long distances.

**Secondary structure features based on AlphaFold models.** Recent advances in Deep Learning techniques enabled de novo modeling of protein structures from sequences with high accuracy that is comparable to experimental methods[50]. Here, we used predicted Alpha-Fold2 models that were downloaded from the resource created by the EMBL Consortium (second release, Date of access: January, 2022)[51], which contains precomputed structures for proteomes of many organisms. Alongside with atomic protein structures, AlphaFold provides the pLDDT score (predicted lDDT-Cα), that is a per-residue

measure of local confidence on a scale from 0 to 100[64]. pLDDT scores were divided into four classes (according to DeepMind classification): [0, 50) - 'very low', [(50,70)] - low', [(70,90)] −'confident', [90, 100] − 'very high'). We used the STRIDE algorithm to annotate the secondary structure based on 3D protein structure[65]. STRIDE assigns one of seven classes to each amino acid in the protein sequence: Alpha helix (H), 3–10 helix (G), PI-helix (I), Extended conformation (E), Isolated bridge (B), Turn (T), Coil (C).

Next, we calculated all possible triads in the form (AA, SSE, pLDDT), where aa belongs to one of 20 types of amino acids, SSE $\ni$ [H, G, I, E, B, T, C], pLDDT $\ni$ ['very low', 'low', 'confident', 'very high']. For longer proteins, AlphaFold2 models consist of overlapping segments of 1400 aa length. In case of discrepancy, when the same amino acid is assigned with different 3D coordinates and pLDDT score, we consolidated the predictions using the following rules: (1) We calculated all possible STRIDE predictions (with different pLDDT scores); (2) we selected the most frequent STRIDE class that had the highest pLDDT score. The total number of features based on structural information provided by AlphaFold2 and STRIDE was 560.

**Gene Ontology features.** Gene Ontology terms are a hierarchal dictionary of annotations describing the function of a particular gene[66,67]. They assign gene characteristics for three directions: molecular function, biological process and cellular component. Each of the direction is represented by a directed acyclic graph, where nodes represent terms (or annotations) and edges represent the relationship of subtype from descendant to ancestor node. There are tens of thousands of terms, therefore one hot encoding for all possible terms is not feasible. To decrease the number of encoded terms, we took only the most frequent terms into account. To estimate the frequency of terms, we used the Swissprot annotations of proteins (after removing redundancy by excluding sequences with more than 30% sequence identity) from Uniprot database[68]. It must be noted, that the frequency of ancestor term $t_a$ also included summation of the frequencies of all descendant terms $t_d$, calculated by the equation:

$$\mathrm{Fr}(t_a) = N_{\mathrm{oc}}(t_a) + \sum_{t_d} N_{\mathrm{oc}}(t_d),$$

where $N_{\mathrm{oc}}(t)$ − number of occurrences of term $t$ in the corpus.

Only terms with frequencies greater than a threshold were kept for feature calculation. Thresholds were chosen for each of the three directions separately to encompass adequate number of terms in the annotations (2500 for molecular function and biological process). We excluded the cellular component direction from the feature selection, as some gene ontology terms contain information about cellular compartments (based on Human Protein Atlas[43], OpenCell[42]). We used one-hot encoding, where each protein was assigned with a vector of fixed length (equal to the number of chosen terms); 1 was assigned if the considered term in the given position (or any of their descendants) was mentioned in protein annotation, otherwise 0. The total number of features describing molecular function and biological process for the set of proteins was 1002.

### Machine learning algorithms

We developed two types of models: one including gene ontology features (PICNIC$_{GO}$) and one without (PICNIC). Both models have the following structure: they consist of 10 Catboost classifiers with dataset for early stopping (to calculate loss function on the dataset different from training to prevent overfitting over training dataset) and fixed depth[69,70]. Catboost is a classifier based on gradient boosting machine (GBM)−a machine learning technique that gives a prediction model in the form of an ensemble of decision trees[71].

Among other gradient boosting classifiers CatBoostClassifier from catboost library showed consistency across multiple runs (we

compared LGBMClassifier from lightgbm library and XGBClassifier from xgboost library). To estimate the overall model performance across multiple runs with different parameters, we used the following metric: we chose the validation score (F1-score) of best iteration of each separate fold, and then computed the mean value across 10 folds.

The model training started with all features, that is 2467 for the model without Gene Ontology features (PICNIC), and 3469 for the model with GO features, $PICNIC_{GO}$. We selected the best features based on feature importance: at each training iteration only features with importance greater than a given threshold were selected for the next run (we took the union of features across different folds). Thus, each subsequent training iteration decreased the number of used features. We chose this feature selection approach because the feature importance did not fluctuate much for different folds (Fig. 1d). The final models contained 18 and 92 features for $PICNIC_{GO}$ and PICNIC, respectively. The number of features in the model with protein annotations are much lesser in comparison with another model as subset of the sequence- and structure-based features connected to condensation properties are already directly encoded by Gene Ontology annotations.

### Cloning, cell culture and imaging
U2OS cells were transfected with plasmids encoding Human proteins tagged with iRFP-670 at their N-terminus. All the 39 tested genes (Dataset S2) were codon optimized (for synthesis ease as well as to override any cellular regulation involving mRNA degradation of the endogenous sequences) and synthesized by Integrated DNA Technologies (IDT), Twist Biosciences and GeneScript (vector maps provided as Dataset S3), restriction digested using NotI-HF and AscI enzymes (NEB-R3189 and NEB-R0558) and then ligated into the pre-digested vectors using T4 DNA ligase(M0202) and transformed in *E. coli* DH5-alpha cells. Positive clones were confirmed by insert release and correctness was verified by DNA sequencing.

Cells were grown in high glucose DMEM medium (Gibco-31966021) supplemented with 10% FBS (Sigma- S0615) as well as 100 units/ml Penicillin-Streptomycin (Gibco-15140112). Following trypsinization with Trypsin-EDTA (Gibco-25300054) cells were seeded on the Ibidi 8 well imaging chamber (80826). After overnight growing the cells, cells were then transfected with plasmids encoding the 24 transgenes and for the colocalization purpose with GFP-tagged DCPa1 (a processing-body marker plasmid) and GFP-tagged Fibrillarin (a sub-nucleolar marker specific to the middle dense fibrillar region allowing the potential dissection of all 3 sub-nucleolar localization of the test proteins) using the Fugene HD transfection reagent (Promega-E2311).

Cellular fluorescent images were recorded on a spinning disk confocal microscope with FRAP capability using the 60x/1.2U-Plan-SApo, water immersion objective lens (Olympus) using 488 nm and 640 nm laser lines, on an Andor-iXon-897-EMCCD camera. High-throughput images were acquired using TDS (technology development studio) at the MPI-CBG, cells were imaged in 96 well plates (Greiner Bio-one F-bottom-655090) using 60x objective lens (UPLSA-PO60xW, NA = 1.2, WD = 0.28mm) with cmos camera (2560x2160 pixels 16 bit) at binning 1 on the automated Yokogawa CV7000 spinning disk microscope using 488 nm and 640 nm laser lines. Image analysis and representative image preparation of the colocalization as well as of the the FRAP movies was done using Fiji[72] (Dataset S4).

Fluorescent microscopy images were quantified manually using Fiji and shape, size descriptors (roundness, perimeter and area) as well as fluorescence intensity were measured (Fig. S9). Enrichment ratio of the condensates was calculated as the ratio of the mean fluorescence of the condensate foci divided by the mean fluorescence of the background. The data was plotted using GraphPad prism 10 software. All the images were finally prepared in Adobe Illustrator. Figure 1b (https://BioRender.com/x62c766) and Fig. S2a (https://BioRender.com/e45y085) were prepared using BioRender: https://BioRender.com/.

### Reporting summary
Further information on research design is available in the Nature Portfolio Reporting Summary linked to this article.

## Data availability
The training, validation and test datasets are available as Dataset S1. The list and properties of the 39 proteins selected for experimental validation is provided as Dataset S2. Plasmid vector maps and representative images are available as Dataset S3, S4. AlphaFold2 structures and images are provided as Dataset S6. The dataset of tested mutations is provided as Dataset S5. All datasets are deposited in the public Edmond repository of the MPG under the link https://doi.org/10.17617/3.0Y9Q8N. Source data are provided with this paper.

## Code availability
The PICNIC code, documentation and examples as jupyter notebooks can be found at https://git.mpi-cbg.de/tothpetroczylab/picnic. Predictions across proteomes of 14 organisms are provided as a web application https://picnic.cd-code.org. PICNIC is also available as a Python package at https://pypi.org/project/picnic-bio/.

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

## Acknowledgements

A.H. was funded by the ELBE postdoctoral fellowship. H.R.S thanks the Nomis foundation for financial support. A.T.-P. acknowledges the Max Planck Gesellschaft MPRGL funding. N.R. was funded by Dewpoint Therapeutics. We would like to thank the Computer Services and Scientific Computing Facilities of the MPI-CBG for their support, especially to Oscar Gonzales for supporting our HPC and HongKee Moon for developing the PICNIC website. We thank Andrei Pozniakovsky for molecular biology support and Natasha Lewis and Barbara Szewczyk for the help with gathering the dataset of mutants. We thank Christina Eugster Oegema from the MPI-CBG central facility (OSCF) for providing routinely Mycoplasma tested and authenticated U2OS cell line used for this study. H.R.S thanks Rico Barsacchi and Martin Stoeter of the Technology of development studio (TDS); Britta Schroth-Diez, Catarina Nabais of the Light microscopy facility (LMF) for the technical support. We are grateful for the support of Michele Marass with manuscript editing and feedback on publication.

## Author contributions

A.H., H.R.S., A.A.H. and A.T-P. designed research; A.H. and H.R.S. performed research; S.G. and N.R. contributed new reagents or analytic tools; M.S. developed the Python package; A.H., H.R.S., S.G. and A.T.P. analyzed data; A.H., H.R.S., A.A.H. and A.T.P. wrote the paper.

## Funding

## Competing interests

A.A.H. is a founder and shareholder of Dewpoint Therapeutics. The remaining authors declare no competing interests.
