## [Transparent Peer Review file · Nature Communications]

PICNIC accurately predicts condensate-forming proteins regardless of their structural disorder across organisms

Corresponding Author: Dr Agnes Toth-Petroczy

Editorial Note: Parts of this Peer Review File have been redacted as indicated to maintain the confidentiality of unpublished data

Version 0:

Reviewer comments:

Reviewer #1

(Remarks to the Author)

Hadarovich and co-authors have developed a new method to predict the likelihood of proteins to take part of biological condensates inside the cell.

The major merits of this paper are related to methods development, as the authors make an effort to go beyond the simple sequence-based predictors of condensation, as well as to train their model using a larger and more rational negative set than previous deep-learning based approaches. I also appreciate the clarity of the methods and how the authors generated a user-friendly database making all of their PICNIC and PICNIC-GO scores available. However, despite a relatively robust method, I think their results fail to teach us much about the biology of condensates. I outline here my concerns. I see the paper as fitting a more specific type of journal (computational methods), but I also suggest some points that can make this more convincing and appealing to a broader audience.

Overall Condensate Biology:

In the introduction author outline examples of condensate function - it follows that - if condensation is functional - proteins that form condensates evolved to do so. I would therefore like to dig deeper on how homologues proteins across organism have such different propensity to form condensates (as reported on the Picnic databases). Why would a protein like SOD1 form condensates in human (0.72), but not in Pombe (0.35) and Drosophila (0.43). Or is the GO part of Picnic confusing the scores here? If so, I suggest thinking twice about how much is it gained by including GOs in the model.

If all proteomes are predicted to contain a 40-50% of sequences able to take part into condensates, what does this tell us about the biology and function of condensates? Are they just a side product of protein sequences - do they happen or do they happen for a reason? What would be the advantage of half of the proteome displaying this ability?

Experimental validation:

- An equal set of proteins predicted not to form condensates should be expressed in the same conditions and evaluated by microscopy
- Evidence that the foci formed by the proteins in Figure 4 are dynamic and reversible should be provided - especially for those forming irregular condensates (KHDC4, TYW5)

Disorder as a predictive feature:

Some sentences regarding protein disorder in the text are contradictory. I understand that the finding that some of their positive hits contain no intrinsically disordered regions shows that disorder is not "essential" for a protein to be part of condensates. However disorder (IUPRED5) is the n.1 scoring feature in their model, so to state (even in the title!) that PICNIC predicts condensate forming proteins regardless of disorder is quite misleading. Different estimates of disordered should be employed in addition to what is shown in Figure 6b (total % disordered residues in each proteome for example) . Likewise "Along with overall sequence complexity and disorder scores of a protein, the secondary structure of individual residue types was also found to be important." Disorder is either important or not important for their prediction and currently it

seems the text wants to overstate the irrelevance of disorder, while it is very relevant - given the results shown.

Method:

I have a concern regarding the risk of circularity in using GO annotations. It is well known that RNA binding proteins are enriched amongst proteins forming or engaging in condensates. Given that the positive sets defined by the authors is based on proteins previously reported to form condensates, it is no wonder that RNA binding comes up as a feature and is then predictive on a held-out testing data. It seems we are not learning anything new from this process.

How powerful is the method in capturing the impact of mutations? There are now tens of examples of mutations altering protein condensation in vitro (and in vivo). It would be good to test the model for its performance on capturing the effect of sequence changes. If the model is accurate and quantitative - it may be helpful, if it's just a classifier for 50% of the proteome I doubt it will be helpful even simply for experimental design.

Reviewer #2

(Remarks to the Author)

Please see the attached pdf file.

Reviewer #3

(Remarks to the Author)

This paper presents PICNIC, a machine learning model to predict proteins involved in biomolecular condensates. The authors have done extensive experimental validation to test their model on 24 proteins, demonstrating its reasonable accuracy. However, I have some major concerns regarding the novelty and utility of the model:

1. The GitHub repository provided by the authors is not very useful in practice. To truly make this a useful tool for the community, the authors should provide code to go from a user-provided protein sequence to generating the AlphaFold models and extracting the required features. Relying on pre-computed databases severely limits the applicability of this method.
2. Since the method relies heavily on databases and features beyond just providing the protein sequence, it is unclear how much improvement is really gained compared to existing predictors (including predictors using GO terms and PPI networks). The novelty and advancements of PICNIC over previous methods should be better highlighted in the Discussion.
3. The predictions on the synuclein family in Fig 3 seem like a black box, as it is unclear which features are driving the ranking. To build confidence, the key features recognized by the model that discriminate between the paralogs should be analyzed and presented. Also, do the 5 AlphaFold models for each synuclein protein agree on the predictions?
4. Fig 6 is confusing with very few data points. To clarify whether there is a significant correlation or not between disorder and predicted condensates, data for more species should be added.

Overall the experimental validation is solid but the novelty is rather limited. Major revisions are needed to address the concerns regarding utility for real applications and clarity around which features are driving the predictions.

Version 2:

Reviewer comments:

Reviewer #1

(Remarks to the Author)

I appreciate the additional work the authors performed by introducing additional metrics, analysis and experiments. Together with the changes to text narrative, I think these make the paper more interesting and more robust.

However, there is one crucial point I disagree on. It is simply not true that the standard practice to validate predictors is to only test the positive predictions. If one wanted to use any metric to evaluate accuracy of PICNIC, such as ROC AUC, PRCs, Matthews correlation coefficient, or even a simple confusion matrix, one would need to know the true negative rate. Given the relatively simple set-up of their validation experiment, I don't see why one would avoid this.

This seems to me even more important in this context, where the authors state that 40-50% of any proteome is predicted to form condensates. To test the accuracy of the method, but also to highlight whether it would truly be a useful tool, it is crucial to know what the background rate of condensation is. I understand it's hard to find a gold standard set of negatives but to claim a conceptual advance, one set of negatives tested in the same conditions as the positives is needed. If, in the

literature, a set of sequences tested in similar conditions exists, then that would also be ok.

If, as the authors state in their rebuttal text, they don't see a point in testing negatives as these sequences could anyway condense under specific circumstance, then what is the point of this - or any other - condensation predictor and what is the meaning of the current validation experiments (performed in just one specific set of conditions)?

(Remarks on code availability)

Reviewer #2

(Remarks to the Author)

The authors adequately addressed my concerns.

(Remarks on code availability)

Reviewer #3

(Remarks to the Author)

The authors have partially enhanced their manuscript, incorporating the database and software package as previously suggested. However, they have not fully addressed some of the concerns raised in earlier comments.

1. The synuclein section's feature importance lacks clarity on whether the features positively or negatively impact predictions. They should consider utilizing SHAP values for clarity, and marking significant feature locations on the protein structure.

2. I remain skeptical about the predictive results in a minority of species being conclusive for the tree of life. The supposed irrelevance might stem from a lack of observed correlation in a limited set of species. I advise more caution in describing the results in Fig.6, avoiding overgeneralized conclusions.

(Remarks on code availability)

Version 3:

Reviewer comments:

Reviewer #1

(Remarks to the Author)

I appreciate the experimental effort the authors undertook and I am glad it has contributed to opening the way for a follow up paper. The lack of "true" negatives should definitely make us all move away from a binary view of condensation. It should also help to define their condition-specific role in cellular fitness, a topic which over the years has been touched upon in different publications with evidence for both condensation-induced cellular toxicity or condensation-induced fitness advantages.

For transparency I recommend including the tested negative set at least in the supplementary file.

I am ok with the rest of the revisions to text and figures and with publication of the manuscript.

(Remarks on code availability)

Reviewer #3

(Remarks to the Author)

The authors have resolved my concerns, and I agree to accept this article.

(Remarks on code availability)

Provisional response to reviewers' comments

Reviewer #1 (Remarks to the Author):

Hadarovich and co-authors have developed a new method to predict the likelihood of proteins to take part of biological condensates inside the cell.

The major merits of this paper are related to methods development, as the authors make an effort to go beyond the simple sequence-based predictors of condensation, as well as to train their model using a larger and more rational negative set than previous deep-learning based approaches. I also appreciate the clarity of the methods and how the authors generated a user-friendly database making all of their PICNIC and PICNIC-GO scores available. However, despite a relatively robust method, I think their results fail to teach us much about the biology of condensates. I outline here my concerns. I see the paper as fitting a more specific type of journal (computational methods), but I also suggest some points that can make this more convincing and appealing to a broader audience.

Overall Condensate Biology:

In the introduction author outline examples of condensate function - it follows that - if condensation is functional - proteins that form condensates evolved to do so. I would therefore like to dig deeper on how homologues proteins across organism have such different propensity to form condensates (as reported on the Picnic databases). Why would a protein like SOD1 form condensates in human (0.72), but not in Pombe (0.35) and Drosophila (0.43). Or is the GO part of Picnic confusing the scores here? If so, I suggest thinking twice about how much is it gained by including GOs in the model.

We agree with the Reviewer, that we would generally expect functional condensation to be conserved across species. For the example that the Reviewer mentioned, the PICNIC scores are consistently high for all species (0.81 for human, 0.79 for Drosophila and 0.80 for *S. pombe*) suggesting a conserved mechanism as expected by the Reviewer. PICNIC-GO scores are indeed not applicable to all species. As we pointed out in the results and discussion, the PICNIC-GO model depends heavily on annotation and for species where annotations are sparse PICNIC-GO score is less reliable. We highlighted it both in the text and on our website that the default score that we recommend to use is PICNIC, without the GO terms.

We acknowledge that PICNIC-GO is biased and not applicable to all cases, therefore, we would be open to remove PICNIC-GO from the manuscript to highlight the advance of PICNIC itself. Our intention, with the inclusion of PICNIC-GO was to show a potential expansion of the tool and it also validated our approach regarding the assembly of the negative dataset.

If all proteomes are predicted to contain a 40-50% of sequences able to take part into condensates, what does this tell us about the biology and function of condensates? Are they just a side product of protein sequences - do they happen or do they happen for a reason? What would be the advantage of half of the proteome displaying this ability?

Indeed, 40-50% of the proteome is a high fraction, and we are also puzzled by the result. It opens new research directions to study protein families that were not yet studied. We surmise that the current literature on condensates is still sparse and biased (for example towards human proteins and proteins containing disordered and low complexity regions). For example, as we showed in the experimental validation, many proteins that do not contain disordered regions can also form condensates. We are also fascinated by the high number of bacterial proteins predicted to form condensates, that is in line with recent discoveries of more and more bacterial condensates. We could only speculate what could be the advantage of having half of the proteome forming condensates. Future research should answer these fascinating questions. Nevertheless, even extensively studied protein have been very recently shown to form condensates in specific conditions, such as hemoglobin in chondrocytes <https://www.nature.com/articles/s41586-023-06611-6>.

Experimental validation:

- An equal set of proteins predicted not to form condensates should be expressed in the same conditions and evaluated by microscopy

The standard practice when validating computational predictors is to test the positive predictions. The main reason is that observing that a protein does not form condensates in a set of experiments cannot rule out its ability to form condensates in, for example, other cell types or other conditions (such as stress). Therefore, proving the negative predictions is more difficult. The same caveat holds for positive predictions, for example not observing condensates in the validation experiment does not rule out that in other conditions it could form condensates. Nevertheless, such experiments provide an upper bound and would only underestimate the performance of our predictor. We would be open, potentially, to perform studies aimed at finding proteins not forming condensates, but since these experiments would not be definitive, we find it more valuable to perform studies on positive results.

We have performed extensive validation on 24 predicted candidate proteins in cellulo. In comparison, previous methods have tested far less of their predictions (see table below).

Table. Experimental validation of phase separation predictors .

Method	Number of positive predictions tested	Number of negative predictions tested	Success	Reference
FuzDrop	2 (in vitro)	None	100%	(Hardenberg et al. 2021)
PSAP	5 (in cellulo)	5 (in cellulo)	100%	(van Mierlo et al. 2021)
DeePhase	None	None	-	(Saar et al. 2021)
PhaSePred	3 (in vitro)	None	100%	(Chen, Z. et al. 2022)

The only work that included negative predictions in their experimental validation was from Mierlo et al., the developers of PSAP predictor. The authors show that for 5 of their negative predictions, have evidence for no condensation, namely those 5 “proteins with a low prediction score displayed a homogeneous GFP signal without clear foci”. But 3 out of 5 these negative proteins are experimentally shown to be members of biological condensates in Human: *CCT2*, <https://cd-code.org/protein/P78371>), *LIMK2* (<https://cd-code.org/protein/P53671>), and *MYO1C* (<https://cd-code.org/protein/O00159>). The above is a good example to illustrate why it is so difficult to trust validation of negative predictions.

In summary, the computational validation on independent benchmarking datasets and the experimental validation agree, and both point towards 0.85 accuracy. Therefore, we do not think that further validation is needed.

- Evidence that the foci formed by the proteins in Figure 4 are dynamic and reversible should be provided - especially for those forming irregular condensates (KHDC4, TYW5)

In order to test if the observed foci are liquid-like and dynamic, we performed FRAP experiments of a few selected proteins. Figure 5c provides such data for 7 proteins including KHDC4. **TO DO: provide experimental data for TYW5 (in progress).**

Disorder as a predictive feature:

Some sentences regarding protein disorder in the text are contradictory. I understand that the finding that some of their positive hits contain no intrinsically disordered regions shows that disorder is not “essential” for a protein to be part of condensates. However disorder (IUPRED5) is the n.1 scoring feature in their model, so to state (even in the title!) that PICNIC predicts condensate forming proteins regardless of disorder is quite misleading. Different estimates of disordered should be employed in

addition to what is shown in Figure 6b (total % disordered residues in each proteome for example) . Likewise “Along with overall sequence complexity and disorder scores of a protein, the secondary structure of individual residue types was also found to be important.” Disorder is either important or not important for their prediction and currently it seems the text wants to overstate the irrelevance of disorder, while it is very relevant - given the results shown.

The overall message that we wished to convey is that disorder is not a prerequisite and not essential for condensation. In contrast, several previous methods assume that disorder is the main determinant of phase separation behavior and thereby bias their results towards identifying mainly disordered proteins. For example, PSAP was trained on a negative dataset of structured proteins (derived from PDB). The authors assumed a priori that structured proteins do not phase-separate. Similarly, FuzDrop uses only disorder scores.

While disorder is indeed important and provides a mechanism for condensate formation via weak and multivalent interactions, it is not the only way. It is not black and white, as the Reviewer suggests: disorder is both important and not important, depending on the combination of other features. The substantial conceptual advance of our method is to consider disorder part of a bigger picture, rather than a “condicio sine qua non” for protein condensation.

We added additional metrics of disorder as suggested by the Reviewer and added 2 more organisms to the analysis (the archaea *I. hospitalis* and *S. islandicus*).

New Supplementary Figure. Overall disorder of a proteome does not correlate with the fraction of predicted condensate forming proteins.

We used three different metrics to quantify disorder of a proteome: A) fraction of proteins with at least one IDR (>40 aa), $R^2=0.03$ (solid line, without grey dots), $R^2=0.17$ (dashed line, all dots), B) Mean of the mean IUPred score of all proteins, $R^2=0.22$ (solid line, without grey dots), $R^2=0.44$ (dashed line, all

dots) C) Mean of the fraction of disordered residues (IUPRED ≥ 0.5) per protein, $R^2=0.30$ (solid line, without grey dots), $R^2=0.34$ (dashed line, all dots).

Method:

I have a concern regarding the risk of circularity in using GO annotations. It is well known that RNA binding proteins are enriched amongst proteins forming or engaging in condensates. Given that the positive sets defined by the authors is based on proteins previously reported to form condensates, it is no wonder that RNA binding comes up as a feature and is then predictive on a held-out testing data. It seems we are not learning anything new from this process.

We agree that PICNIC-GO is biased towards more annotated proteins and we have highlighted it ourselves in the text of the paper (page 16, 'Although gene ontology terms are valuable resources of information, they could introduce bias due to their nature of annotation') and on the website of picnic.cd-code.org.

We emphasize that the paper is focused on the model PICNIC that doesn't use gene ontology annotation and should be used by default. PICNIC-GO is heavily dependent on annotation, nevertheless it can help in cases of well-annotated proteins that are enriched in the properties that are leading to condensate formation, but are not experimentally validated yet.

The reviewer is right, that we did not learn new biology here. Nevertheless, seeing RNA binding as a top feature served as a sanity check. We were pleased to recover a known phenomena, thereby validating our machine learning approach (the design of our training datasets).

How powerful is the method in capturing the impact of mutations? There are now tens of examples of mutations altering protein condensation in vitro (and in vivo). It would be good to test the model for its performance on capturing the effect of sequence changes. If the model is accurate and quantitative - it may be helpful, if it's just a classifier for 50% of the proteome I doubt it will be helpful even simply for experimental design.

Capturing the impact of single mutations is a real challenge in various fields of computational biology, including structure prediction (AlphaFold2 is insensitive to mutations) and phase separation prediction. The example about the synuclein family shows that our method is able to predict the condensate-formation within paralogs with high sequence similarity.

We do not expect our method to be sensitive enough for the effects of single mutations, as none of the other tools for condensate predictions do. We acknowledge that this is a very interesting and thought-provoking question, and we hope that our approach will facilitate the development of tools aimed at addressing it. **TO DO: We will add this caveat to the discussion.**

Although PICNIC is not able to predict the effect of a mutation, extracting the most important features for each individual protein of the algorithm can give a hint about properties, specific for this particular protein, that are important for its condensation. Here is the example for FUS protein where the most important features are listed (New Supplementary Figure). The features highlighted by PICNIC for FUS protein condensation are in agreement with a study which shows that phase separation is primarily governed by multivalent interactions among tyrosine residues from prion-like domains and arginine residues from RNA-binding domains, which are modulated by negatively charged residues. Glycine residues enhance the fluidity, whereas glutamine and serine residues promote hardening (Wang, J. et al, 2018). Thus, analysing PICNIC features can generate hypothesis for potential mutations that may alter condensation. **To DO: This possibility can be added in the next version of the PICNIC software.**

New Supplementary Figure. The top features picked by PICNIC to predict FUS (Uniprot ID: FUS_HUMAN) condensation.

Reviewer #2 (Remarks to the Author):

Comments on “PICNIC accurately predicts condensate-forming proteins regardless of their structural disorder across organisms” by Toth-Petroczy and colleagues In this manuscript, the authors are developing a new machine learning program to predict potential condensate-forming proteins in cells. What they did first was to define a positive and a negative dataset for training and testing purposes. They smartly used a minus-strategy to establish the negative dataset by eliminating all proteins that were known to interact with known condensate members. Their classifier, called PICNIC (Proteins Involved in CoNDensates In Cell), is based on sequence-distance-based and structure-based features derived from AlphaFold2 models. To further increase the performance of their model, they integrated functional information that is already known about each protein and described as Gene Ontology (GO) terms. The second classifier is called PICNICGO that combines GO terms and the previously used PICNIC features. Overall, PICNIC and PICNICGO outperform existing programs for identifying proteins involved in biomolecular condensate formation. Using α -, β - and -synuclein as a testing case, PICNIC is the only program that correctly predicted their respective propensity in condensate-involvement. They used 24 proteins to test their capacity to form condensates upon overexpression in cellulose. Among them, 21 indeed formed condensates. Last but not the least, the authors found out that Proteome-wide predictions detect no correlation of predicted condensate proteome size with disorder content and organismal complexity. In my opinion, PICNIC (and PICNICGO) is a useful program for the field of biomolecular condensate to have. Many researchers will benefit from it. Nevertheless, I hope that the authors can address my two critiques below.

Major critiques: 1) Notably PICNICGO utilizes Gene Ontology (GO) analysis to infer phase separation potential through the lens of known protein functionalities, setting itself apart from existing prediction tools. Nevertheless, it tends to exhibit certain degree of bias towards proteins with precisely defined functions.

We agree that PICNIC-GO is biased towards more annotated proteins and are highlighting it in the text of the paper (page 16, ‘Although gene ontology terms are valuable resources of information, they could introduce bias due to their nature of annotation’). We emphasize that the paper is focused on the model PICNIC that doesn’t use gene ontology annotation and should be used by default. PICNIC-GO is heavily dependent on annotation, nevertheless it can help in cases of well-annotated proteins that are enriched in the properties that are leading to condensation formation, but are not experimentally validated yet. Based on the comments of Reviewer 1, we decided to remove PICNIC-GO from the results section and only present it in the discussion as a minor proof-of concept extension of PICNIC.

Therefore, we recommend incorporating insights from existing imaging data to even further refine the ultimate output assessments.

We agree that additional experimental data can help in improving prediction. For example, one of the models of a meta-predictor (<http://predict.phasep.pro/>) uses microscopy data as one of the input features for the prediction (note: for human data only). The bottleneck of these approaches is that these data are limited and biased towards certain proteins. Another problem is availability of such data for other organisms, which makes it difficult to make the models generalizable across different species.

2) Speaking of bias, the program is good at scoring proteins that are known to form condensates, of which many might be in their training positive dataset. However, it scores less optimally on unpublished proteins that are capable of forming condensates. Examples for both classes of proteins and their corresponding PICNIC and PICNICGO scores are shown below. I hope that the authors can address this bias by improving the program a bit or at least give an explanation of this bias.
(table)

For all supervised methods there is a bias to predict instances that are part of the training set with higher accuracy. This is the case for PICNIC as well as the Reviewer suggests. Since 9 out of 13 “published” proteins from the Table assembled by the Reviewer are present in the training dataset, they are predicted with high (100%) accuracy by PICNIC. While for the “unpublished dataset” only 4 out of 7 are predicted correctly (~57% accuracy). Interestingly, 2 out of the 7 proteins (marked as unpublished) are also present in the training dataset: PITX2 was a part of first release of PhaSepDB as high-throughput data, and FUBP1 is a part of the Stress granule (<https://cd-code.org/protein/Q96AE4>). It is worth mentioning that although PITX2 is a part of the training dataset,

the score is still below the threshold of 0.5, as a machine learning approach does not directly translate all training dataset to the positive scores.

Reviewer #3 (Remarks to the Author):

This paper presents PICNIC, a machine learning model to predict proteins involved in biomolecular condensates. The authors have done extensive experimental validation to test their model on 24 proteins, demonstrating its reasonable accuracy. However, I have some major concerns regarding the novelty and utility of the model:

1. The GitHub repository provided by the authors is not very useful in practice. To truly make this a useful tool for the community, the authors should provide code to go from a user-provided protein sequence to generating the AlphaFold models and extracting the required features.

Additionally to the GitLab repository, now we also provide the software as a pip installable python package (picnic-bio, <https://pypi.org/project/picnic-bio/1.0.0b1/>) for easier distribution and usage for the biomedical community. We would like to add a new co-author, Maxim Scheremtjew, who implemented the software.

We provide users the availability to run PICNIC from the command line in two modes: automatic and manual (<https://git.mpi-cbg.de/tothpetroczylab/picnic>).

The automatic mode works for proteins with length < 1400 aa, with precalculated AlphaFold2 models deposited to UniprotKB (it will be automatically fetched from the database). The manual mode requires uniprot_id, AlphaFold2 model(s) and fasta file with sequence to be provided as input.

The users can use other (web)services or their own AlphaFold2 installations to generate the AF models and then input those to PICNIC. The reason why we do not calculate the AF models is, that structure prediction by AlphaFold2 model is the most computationally expensive task (needs GPUs and can take hours for large proteins). We wanted to decouple the PICNIC software from these heavy computational requirements and we aimed at providing a fast service to predict proteins' ability to be part of biomolecular condensate.

Relying on pre-computed databases severely limits the applicability of this method.

We agree that relying on pre-computed databases severely limits the applicability of this method. Additionally to the public gitlab repository that we shared for the initial submission, we are now providing a pip-installable package, called bio_picnic (see screenshot below). Further, we are working on developing a web-service that would allow users to input the protein (by uniprot id) and provide calculation. Still we cannot support the calculation of AlphaFold models due to high computational costs, therefore the users will have to provide those (compute using their own or other resources) or use precomputed structures, that is pipeline cat automatically fetch (from uniprot).

2. Since the method relies heavily on databases and features beyond just providing the protein sequence, it is unclear how much improvement is really gained compared to existing predictors (including predictors using GO terms and PPI networks).

The data quality is of utter importance for all algorithms based on machine learning approaches. The PICNIC algorithm utilizes only protein sequence information and structural models based on sequence input built with AlphaFold2 algorithm. Figure 2 shows the performance gain to other sequence based predictors on 3 different validation datasets.

We would like to stress, that only the extended method, PICNICGO algorithm, uses in addition Gene Ontology information retrieved from UniprotKB. As more information becomes available, more precise prediction becomes possible. Many predictors make use of cleaner and more annotated corpus of data, thus improving the accuracy of predictions. For example, the recently published meta-predictor (link, <http://predict.phasep.pro/>) combines 8 scores provided by other algorithms for one model and 10 scores (including experimental information) for another model. Nevertheless, this model is also biased by the availability of microscopy data and limited to human proteins only.

The novelty and advancements of PICNIC over previous methods should be better highlighted in the Discussion.

TO DO: Rewrite discussion.

3. The predictions on the synuclein family in Fig 3 seem like a black box, as it is unclear which features are driving the ranking. To build confidence, the key features recognized by the model that discriminate between the paralogs should be analyzed and presented.

The comment above is applicable to many supervised methods (including PSAP, DeePhase). Nevertheless, we agree that it is important to look for features that contribute the most to the predictions.

The plot below shows the features that contributed the most to the prediction for each synuclein protein. We can see that features that stand out in beta-synuclein and are absent in alpha and gamma (I-AlphaHelix-I, F-AlphaHelix-I) are connected to hydrophobic amino acids, being part of alpha-helix with low pLDDT score. Thus, the structural changes involving the alpha-helix that we also mentioned previously, are likely to drive the signal.

Also, do the 5 AlphaFold models for each synuclein protein agree on the predictions?

We took the models from UniprotKB, that provides only one (best ranked) model per protein, and did not save all 5 models.

Overall, because the pLDDT score is low for the variable part of the structure for beta synuclein, it is likely that there are differences in the 5 models (hence the low pLLDT score). However, that is exactly one of the features for prediction, i.e. AF is less sure about the model, because the composition is different and probably contradicts and doesn't allow AF to build alpha-helix with high reliability - that feature was shown to be important while estimating protein ability to condensate.

4. Fig 6 is confusing with very few data points. To clarify whether there is a significant correlation or not between disorder and predicted condensates, data for more species should be added.

We added additional metrics of disorder as suggested by the Reviewer and added 2 more organisms to the analysis (the archaea *I. hospitalis* and *S. islandicus*).

New Supplementary Figure. Overall disorder of a proteome does not correlate with the fraction of predicted condensate forming proteins. We used three different metrics to quantify disorder of a proteome: A) fraction of proteins with at least one IDR (>40 aa), $R^2=0.03$ (solid line, without grey dots), $R^2=0.17$ (dashed line, all dots), B) Mean of the mean IUPred score of all proteins, $R^2=0.22$ (solid line, without grey dots), $R^2=0.44$ (dashed line, all

dots) C) Mean of the fraction of disordered residues (IUPRED ≥ 0.5) per protein, $R^2=0.30$ (solid line, without grey dots), $R^2=0.34$ (dashed line, all dots).

Overall the experimental validation is solid but the novelty is rather limited. Major revisions are needed to address the concerns regarding utility for real applications and clarity around which features are driving the predictions.

Response to reviewers' comments

Reviewer #1 (Remarks to the Author):

Hadarovich and co-authors have developed a new method to predict the likelihood of proteins to take part of biological condensates inside the cell.

The major merits of this paper are related to methods development, as the authors make an effort to go beyond the simple sequence-based predictors of condensation, as well as to train their model using a larger and more rational negative set than previous deep-learning based approaches. I also appreciate the clarity of the methods and how the authors generated a user-friendly database making all of their PICNIC and PICNIC-GO scores available. However, despite a relatively robust method, I think their results fail to teach us much about the biology of condensates. I outline here my concerns. I see the paper as fitting a more specific type of journal (computational methods), but I also suggest some points that can make this more convincing and appealing to a broader audience.

Overall Condensate Biology:

In the introduction author outline examples of condensate function - it follows that - if condensation is functional - proteins that form condensates evolved to do so. I would therefore like to dig deeper on how homologues proteins across organism have such different propensity to form condensates (as reported on the Picnic databases). Why would a protein like SOD1 form condensates in human (0.72), but not in Pombe (0.35) and Drosophila (0.43). Or is the GO part of Picnic confusing the scores here? If so, I suggest thinking twice about how much is it gained by including GOs in the model.

We agree with the Reviewer, that we would generally expect functional condensation to be conserved across species. For the example that the Reviewer mentioned, the PICNIC scores are consistently high for all species (0.81 for human, 0.79 for Drosophila and 0.80 for *S. pombe*) suggesting a conserved mechanism as expected by the Reviewer. PICNIC-GO scores are indeed not applicable to all species. As we pointed out in the results and discussion, the PICNIC-GO model depends heavily on annotation and for species where annotations are sparse PICNIC-GO score is less reliable. We highlighted it both in the text and on our website that the default score that we recommend to use is PICNIC, without the GO terms.

We acknowledge that PICNIC-GO is biased and not applicable to all cases, therefore, we would be open to remove PICNIC-GO from the manuscript to highlight the advance of PICNIC itself. Our intention, with the inclusion of PICNIC-GO was to show a potential expansion of the tool and it also validated our approach regarding the assembly of the negative dataset.

If all proteomes are predicted to contain a 40-50% of sequences able to take part into condensates, what does this tell us about the biology and function of condensates? Are they just a side product of protein sequences - do they happen or do they happen for a reason? What would be the advantage of half of the proteome displaying this ability?

Indeed, 40-50% of the proteome is a high fraction, and we are also puzzled by the result. It opens new research directions to study protein families that were not yet studied. We surmise that the current literature on condensates is still sparse and biased (for example towards human proteins and proteins containing disordered and low complexity regions). For example, as we showed in the experimental validation, many proteins that do not contain disordered regions can also form condensates. We are also fascinated by the high number of bacterial proteins predicted to form condensates, that is in line with recent discoveries of more and more bacterial condensates. We could only speculate what could be the advantage of having half of the proteome forming condensates. Future research should answer these fascinating questions. Nevertheless, even extensively studied protein have been very recently shown to form condensates in specific conditions, such as hemoglobin in chondrocytes <https://www.nature.com/articles/s41586-023-06611-6>.

Experimental validation:

- An equal set of proteins predicted not to form condensates should be expressed in the same conditions and evaluated by microscopy

The standard practice when validating computational predictors is to test the positive predictions. The main reason is that observing that a protein does not form condensates in a set of experiments cannot rule out its ability to form condensates in, for example, other cell types or other conditions (such as stress). Therefore, proving the negative predictions is more difficult. The same caveat holds for positive predictions, for example not observing condensates in the validation experiment does not rule out that in other conditions it could form condensates. Nevertheless, such experiments provide an upper bound and would only underestimate the performance of our predictor. We would be open, potentially, to perform studies aimed at finding proteins not forming condensates, but since these experiments would not be definitive, we find it more valuable to perform studies on positive results.

We have performed extensive validation on 24 predicted candidate proteins in cellulo. In comparison, previous methods have tested far less of their predictions (see table below).

Table. Experimental validation of phase separation predictors .

Method	Number of positive predictions tested	Number of negative predictions tested	Success	Reference
FuzDrop	2 (in vitro)	None	100%	(Hardenberg et al. 2021)
PSAP	5 (in cellulo)	5 (in cellulo)*	100%	(van Mierlo et al. 2021)
DeePhase	None	None	-	(Saar et al. 2021)
PhaSePred	3 (in vitro)	None	100%	(Chen, Z. et al. 2022)

*The only work that included negative predictions in their experimental validation was from Mierlo et al., the developers of PSAP predictor. The authors show that for 5 of their negative predictions, have evidence for no condensation, namely those 5 “proteins with a low prediction score displayed a homogeneous GFP signal without clear foci”. **But 3 out of 5 these negative proteins are experimentally shown to be members of biological condensates in Human: CCT2, <https://cd-code.org/protein/P78371>, LIMK2 (<https://cd-code.org/protein/P53671>), and MYO1C (<https://cd-code.org/protein/O00159>).** The above is a good example to illustrate why it is so difficult to trust validation of negative predictions.

In summary, the computational validation on independent benchmarking datasets and the experimental validation agree, and both point towards 0.85 accuracy. Therefore, we do not think that further validation is needed.

- Evidence that the foci formed by the proteins in Figure 4 are dynamic and reversible should be provided - especially for those forming irregular condensates (KHDC4, TYW5)

In order to test if the observed foci are liquid-like and dynamic, we performed FRAP experiments of a few selected proteins. The previous version of Figure 5c provided such data for 7 proteins including KHDC4. Now we also included FRAP experiments about TYW5. Please see the updated figure below.

6 out of 8 tested proteins show very fast dynamics (at a time scale of 20 Seconds, indicated) indicative of liquid-like nature, while two (RBM1D and TYW5) show no recovery at the indicated time.

Representative FRAP recovery profiles of the condensate localising proteins

Additionally, we provide now 10 representative images of cells expressing all 24 proteins tested as Suppl. Fig.

Figure S10. Gallery of cells expressing the 24 proteins tested.

For each protein tested 10 different images showing one to many cells in the same field of view consistently demonstrating the presence of condensates for 21 out of 24 proteins Interestingly, some proteins showed localization to more than one type/site of condensate. For example, DRC4: nuclear and cytoplasmic; MRPL1: nuclear and cytoplasmic bodies; RS10L: cytoplasmic bodies/ filaments as well as nucleolar localization.

Disorder as a predictive feature:

Some sentences regarding protein disorder in the text are contradictory. I understand that the finding that some of their positive hits contain no intrinsically disordered regions shows that disorder is not

“essential” for a protein to be part of condensates. However disorder (IUPRED5) is the n.1 scoring feature in their model, so to state (even in the title!) that PICNIC predicts condensate forming proteins regardless of disorder is quite misleading. Different estimates of disordered should be employed in addition to what is shown in Figure 6b (total % disordered residues in each proteome for example) . Likewise “Along with overall sequence complexity and disorder scores of a protein, the secondary structure of individual residue types was also found to be important.” Disorder is either important or not important for their prediction and currently it seems the text wants to overstate the irrelevance of disorder, while it is very relevant - given the results shown.

The overall message that we wished to convey is that disorder is not a prerequisite and not essential for condensation. In contrast, several previous methods assume that disorder is the main determinant of phase separation behavior and thereby bias their results towards identifying mainly disordered proteins. For example, PSAP was trained on a negative dataset of structured proteins (derived from PDB). The authors assumed a priori that structured proteins do not phase-separate. Similarly, FuzDrop uses only disorder scores.

While disorder is indeed important and provides a mechanism for condensate formation via weak and multivalent interactions, it is not the only way. It is not black and white, as the Reviewer suggests: disorder is both important and not important, depending on the combination of other features. The substantial conceptual advance of our method is to consider disorder part of a bigger picture, rather than a “condicio sine qua non” for protein condensation.

We added additional metrics of disorder as suggested by the Reviewer and added 2 more organisms to the analysis (the archaea *I. hospitalis* and *S. islandicus*).

New Supplementary Figure S11. Overall disorder of a proteome does not correlate with the fraction of predicted condensate forming proteins.

We used three different metrics to quantify disorder of a proteome: A) fraction of proteins with at least one IDR (>40 aa), $R^2=0.03$ (solid line, without grey dots), $R^2=0.17$ (dashed line, all dots), B) Mean of the mean IUPred score of all proteins, $R^2=0.22$ (solid line, without grey dots), $R^2=0.44$ (dashed line, all

dots) C) Mean of the fraction of disordered residues (IUPRED ≥ 0.5) per protein, $R^2=0.30$ (solid line, without grey dots), $R^2=0.34$ (dashed line, all dots).

Method:

I have a concern regarding the risk of circularity in using GO annotations. It is well known that RNA binding proteins are enriched amongst proteins forming or engaging in condensates. Given that the positive sets defined by the authors is based on proteins previously reported to form condensates, it is no wonder that RNA binding comes up as a feature and is then predictive on a held-out testing data. It seems we are not learning anything new from this process.

We agree that PICNIC-GO is biased towards more annotated proteins and we have highlighted it ourselves in the text of the paper (page 16, 'Although gene ontology terms are valuable resources of information, they could introduce bias due to their nature of annotation') and on the website of picnic.cd-code.org.

We emphasize that the paper is focused on the model PICNIC that doesn't use gene ontology annotation and should be used by default. PICNIC-GO is heavily dependent on annotation, nevertheless it can help in cases of well-annotated proteins that are enriched in the properties that are leading to condensate formation, but are not experimentally validated yet.

The reviewer is right, that we did not learn new biology here. Nevertheless, seeing RNA binding as a top feature served as a sanity check. We were pleased to recover a known phenomena, thereby validating our machine learning approach (e.g. the design of our training datasets).

How powerful is the method in capturing the impact of mutations? There are now tens of examples of mutations altering protein condensation *in vitro* (and *in vivo*). It would be good to test the model for its performance on capturing the effect of sequence changes. If the model is accurate and quantitative - it may be helpful, if it's just a classifier for 50% of the proteome I doubt it will be helpful even simply for experimental design.

Capturing the impact of single mutations is a real challenge in various fields of computational biology, including structure prediction (AlphaFold2 is insensitive to mutations) and phase separation prediction. The example about the synuclein family shows that our method is able to predict the condensate-formation within paralogs with high sequence similarity.

We do not expect our method to be sensitive enough for the effects of single mutations, as none of the other tools for condensate predictions do.

Nevertheless, we assembled a set of sequences that contain single mutations, multiple mutations and deletions that reduce or abolish the condensate formation of a protein (Table S1). We were pleased to see that PICNIC scores were consistently lower for mutants that lead to reduced condensates formation (New Figure 3e, red or yellow) than the corresponding WT sequence (New Figure 3e green stars).

New Figure 3. PICNIC captures the different phase separation behaviour of paralogs and mutant sequences. **a)** The three paralogs in human share high sequence identity as depicted in the multiple sequence alignment. **b)** Structural models for α -synuclein (yellow), β -synuclein (cyan) and γ -synuclein (green), predicted by AlphaFold2 reveal that β -synuclein has a bent structure. **c)** Despite the high sequence similarity, only α - and γ -synuclein are part of biomolecular condensates, while β -synuclein has not been found in any biomolecular condensates yet and was shown not to phase separate *in vitro*. **d)** Comparison of prediction scores of different tools in identifying condensate forming (α and γ , green) and non-condensate

forming paralog (β , red). PICNIC accurately predicts the condensate-forming ability of the synuclein family, and ranks β -synuclein the lowest, while other tools give equivalent scores to all paralogs or fail to identify the right trend. Vertical lines indicate the threshold used by the various methods to classify condensate-forming proteins. e) PICNIC scores of WT (shown as stars) and mutant sequences assembled from the literature (Table S1). f) Example of PICNIC's performance on the mutated sequences of CBX2. Whereas the scores for the canonical sequence and mutant CBX2_DEA, that both form condensates (green stars) are high, the score decreases for the mutants with reduced ability to condensate (empty red stars) (CBX2_10) and (CBX2_16), and for the mutants CBX2_13 and CBX2_23 that do not form condensates (red stars). Sequence alignment of the canonical sequence of CBX2 (green) and the mutated sequences studied here. On the left panel the structural alignment between CBX2 (green) and CBX2_13 (red), as well as CBX2 (green) and CBX2_23 (purple) points out that even with preserved SSE, their 3D orientation affects the proteins' property to condensate.

We would like to note, that none of the other tools that we benchmarked against could be applied off-the-shelf to compute scores for mutated sequences. Specifically, PhaSepRed, PSAP and X only provide scores for Uniprot IDs as inputs, i.e. WT sequences. Thus, PICNIC is the first method that provides predictions for any sequence, mutated or synthetic.

Additionally, extracting the most important features for each individual protein of the algorithm can give a hint about properties, specific for this particular protein, that are important for its condensation. Here is the example for FUS protein where the most important features are listed (**New Supplementary Figure 12**). The features highlighted by PICNIC for FUS protein condensation are in agreement with a study which shows that phase separation is primarily governed by multivalent interactions among tyrosine residues from prion-like domains and arginine residues from RNA-binding domains, which are modulated by negatively charged residues. Glycine residues enhance the fluidity, whereas glutamine and serine residues promote hardening (Wang, J. et al, 2018). Thus, analysing PICNIC features can generate hypothesis for potential mutations that may alter condensation.

New Supplementary Figure S12. The top features picked by PICNIC to predict FUS (Uniprot ID: FUS_HUMAN) condensation.

Overall, our results about mutation effects and the feature analysis suggest, that PICNIC can facilitate the experimental design of sequences that may abolish condensate formation or vica versa, could generate hypothesis what sequence changes may promote condensate formation.

Reviewer #2 (Remarks to the Author):

Comments on “PICNIC accurately predicts condensate-forming proteins regardless of their structural disorder across organisms” by Toth-Petroczy and colleagues In this manuscript, the authors are developing a new machine learning program to predict potential condensate-forming proteins in cells. What they did first was to define a positive and a negative dataset for training and testing purposes. They smartly used a minus-strategy to establish the negative dataset by eliminating all proteins that were known to interact with known condensate members. Their classifier, called PICNIC (Proteins Involved in CoNDensates In Cell), is based on sequence-distance-based and structure-based features derived from AlphaFold2 models. To further increase the performance of their model, they integrated functional information that is already known about each protein and described as Gene Ontology (GO) terms. The second classifier is called PICNICGO that combines GO terms and the previously used PICNIC features. Overall, PICNIC and PICNICGO outperform existing programs for identifying proteins involved in biomolecular condensate formation. Using α -, β - and -synuclein as a testing case, PICNIC is the only program that correctly predicted their respective propensity in condensate-involvement. They used 24 proteins to test their capacity to form condensates upon overexpression in cellulose. Among them, 21 indeed formed condensates. Last but not the least, the authors found out that Proteome-wide predictions detect no correlation of predicted condensate proteome size with disorder content and organismal complexity. In my opinion, PICNIC (and PICNICGO) is a useful program for the field of biomolecular condensate to have. Many researchers will benefit from it. Nevertheless, I hope that the authors can address my two critiques below.

Major critiques: 1) Notably PICNICGO utilizes Gene Ontology (GO) analysis to infer phase separation potential through the lens of known protein functionalities, setting itself apart from existing prediction tools. Nevertheless, it tends to exhibit certain degree of bias towards proteins with precisely defined functions.

We agree that PICNIC-GO is biased towards more annotated proteins and are highlighting it in the text of the paper (page 16, ‘Although gene ontology terms are valuable resources of information, they could introduce bias due to their nature of annotation’). We emphasize that the paper is focused on the PICNIC model that doesn’t use gene ontology annotation and should be used by default. PICNIC-GO is heavily dependent on annotation, nevertheless it can help in cases of well-annotated proteins that are enriched in the properties that are leading to condensation formation, but are not experimentally validated yet.

Based on the comments of Reviewer 1, we decided to remove PICNIC-GO from the results section and only present it in the discussion as a minor proof-of concept extension of PICNIC.

Therefore, we recommend incorporating insights from existing imaging data to even further refine the ultimate output assessments.

We agree that additional experimental data can help in improving prediction. For example, one of the models of a meta-predictor (<http://predict.phasep.pro/>) uses microscopy data as one of the input features for the prediction (note: for human data only). The bottleneck of these approaches is that these data are limited and biased towards certain proteins. Another problem is availability of such data for other organisms, which makes it difficult to make the models generalizable across different species.

2) Speaking of bias, the program is good at scoring proteins that are known to form condensates, of which many might be in their training positive dataset. However, it scores less optimally on unpublished proteins that are capable of forming condensates. Examples for both classes of proteins and their corresponding PICNIC and PICNICGO scores are shown below. I hope that the authors can address this bias by improving the program a bit or at least give an explanation of this bias.
(table)

The reviewer is right. For all supervised methods there is a bias to predict instances that are part of the training set with higher accuracy. This is the case for PICNIC as well as the Reviewer suggests. Since 9 out of 13 “published” proteins from the Table assembled by the Reviewer are present in the training dataset, they are predicted with high (100%) accuracy by PICNIC. While for the “unpublished dataset” only 4 out of 7 are predicted correctly (~57% accuracy). Interestingly, 2 out of the 7 proteins (marked as unpublished) are also present in the training dataset: PITX2 was a part of first release of PhaSepDB as high-throughput data, and FUBP1 is a part of the Stress granule (<https://cd->

[code.org/protein/Q96AE4](https://www.uniprot.org/protein/Q96AE4)). It is worth mentioning that although PITX2 is a part of the training dataset, the score is still below the threshold of 0.5, as a machine learning approach does not directly translate all training dataset to the positive scores.

Reviewer #3 (Remarks to the Author):

This paper presents PICNIC, a machine learning model to predict proteins involved in biomolecular condensates. The authors have done extensive experimental validation to test their model on 24 proteins, demonstrating its reasonable accuracy. However, I have some major concerns regarding the novelty and utility of the model:

1. The GitHub repository provided by the authors is not very useful in practice. To truly make this a useful tool for the community, the authors should provide code to go from a user-provided protein sequence to generating the AlphaFold models and extracting the required features.

Additionally to the GitLab repository, now we also provide the software as a pip installable python package (picnic-bio, <https://pypi.org/project/picnic-bio/1.0.0b1/>) for easier distribution and usage for the biomedical community. We would like to add a new co-author, Maxim Scheremtjew, who implemented the software.

We provide users the availability to run PICNIC from the command line in two modes: automatic and manual (<https://git.mpi-cbg.de/tothpetroczylab/picnic>).

The automatic mode works for proteins with length < 1400 aa, with precalculated Alphafold2 models deposited to UniprotKB (it will be automatically fetched from the database). The manual mode requires uniprot_id, Alphafold2 model(s) and fasta file with sequence to be provided as input.

The users can use other (web)services or their own AlphaFold2 installations to generate the AF models and then input those to PICNIC. The reason why we do not calculate the AF models is, that structure prediction by Alphafold2 model is the most computationally expensive task (needs GPUs and can take hours for large proteins). We wanted to decouple the PICNIC software from these heavy computational requirements and we aimed at providing a fast service to predict proteins' ability to be part of biomolecular condensate.

Relying on pre-computed databases severely limits the applicability of this method.

We agree that relying on pre-computed databases severely limits the applicability of this method. Additionally to the public gitlab repository that we shared for the initial submission, we are now providing a pip-installable package, called bio_picnic (see screenshot below). Further, we are working on developing a web-service that would allow users to input the protein (by uniprot id) and provide calculation. Still we cannot support the calculation of AlphaFold models due to high computational costs, therefore the users will have to provide those (compute using their own or other resources) or use precomputed structures, that is pipeline cat automatically fetch (from uniprot).

2. Since the method relies heavily on databases and features beyond just providing the protein sequence, it is unclear how much improvement is really gained compared to existing predictors (including predictors using GO terms and PPI networks).

The data quality is of utter importance for all algorithms based on machine learning approaches. The PICNIC algorithm utilizes only protein sequence information and structural models based on sequence input built with Alphafold2 algorithm. Figure 2 shows the performance gain to other sequence based predictors on 3 different validation datasets.

We would like to stress, that only the extended method, PICNIC-GO algorithm, uses in addition Gene Ontology information retrieved from UniprotKB. As more information becomes available, more precise prediction becomes possible. Many predictors make use of cleaner and more annotated corpus of data, thus improving the accuracy of predictions. For example, the recently published meta-predictor (link, <http://predict.phasep.pro/>) combines 8 scores provided by other algorithms for one model and 10 scores (including experimental information) for another model. Nevertheless, this model is also biased by the availability of microscopy data and limited to human proteins only.

The novelty and advancements of PICNIC over previous methods should be better highlighted in the Discussion.

We expanded the discussion now and added a paragraph about the performance on mutated sequences.

3. The predictions on the synuclein family in Fig 3 seem like a black box, as it is unclear which features are driving the ranking. To build confidence, the key features recognized by the model that discriminate between the paralogs should be analyzed and presented.

The comment above is applicable to many supervised methods (including PSAP, DeePhase). Nevertheless, we agree that it is important to look for features that contribute the most to the predictions.

The plot below shows the features that contributed the most to the prediction for each synuclein protein. We can see that features that stand out in beta-synuclein and are absent in alpha and gamma (I-AlphaHelix-I, F-AlphaHelix-I) are connected to hydrophobic amino acids, being part of alpha-helix with low plddt score. Thus, the structural changes involving the alpha-helix that we also mentioned previously, are likely to drive the signal.

Thank you for this suggestion, we added this analysis to the manuscript.

New Supplementary Figure S13. The top features picked by PICNIC regarding the synuclein protein family: alpha, beta and gamma-synuclein.

Also, do the 5 AlphaFold models for each synuclein protein agree on the predictions?

We took the models from UniprotKB, that provides only one (best ranked) model per protein, and did not save all 5 models.

Overall, because the pLDDT score is low for the variable part of the structure for beta synuclein, it is likely that there are differences in the 5 models (hence the low pLLDT score). However, that is exactly one of the features for prediction, i.e. AF is less sure about the model, because the composition is different and probably contradicts and doesn't allow AF to build alpha-helix with high reliability - that feature was shown to be important while estimating the protein's ability to condensate.

4. Fig 6 is confusing with very few data points. To clarify whether there is a significant correlation or not between disorder and predicted condensates, data for more species should be added.

We added additional metrics of disorder as suggested by the Reviewer and added 2 more organisms to the analysis (the archaea *I. hospitalis* and *S. islandicus*). And our conclusions remained, that we

see no correlation with disorder content of a proteome with the fraction of proteins predicted to form condensates. We also think, that these results are surprising.

New Supplementary Figure S11. Overall disorder of a proteome does not correlate with the fraction of predicted condensate forming proteins.

We used three different metrics to quantify disorder of a proteome: A) fraction of proteins with at least one IDR (>40 aa), $R^2=0.03$ (solid line, without grey dots), $R^2=0.17$ (dashed line, all dots), B) Mean of the mean IUPred score of all proteins, $R^2=0.22$ (solid line, without grey dots), $R^2=0.44$ (dashed line, all dots) C) Mean of the fraction of disordered residues (IUPRED ≥ 0.5) per protein, $R^2=0.30$ (solid line, without grey dots), $R^2=0.34$ (dashed line, all dots).

Overall the experimental validation is solid but the novelty is rather limited. Major revisions are needed to address the concerns regarding utility for real applications and clarity around which features are driving the predictions.

The success of our approach is due to several innovations:

1. As amino acid composition bias and patterning of charges were shown to impact the ability of proteins to form condensates⁵⁻⁷, we developed features that represent short and long range co-occurrences of amino acids in the protein sequence and structure.
2. We used the largest curated dataset of known biomolecular condensates (CD-CODE.org, 2142 non redundant proteins) recently published by our lab⁸. Whereas PSAP used only 90, DeePhase 137, FuzDrop 453 proteins as positive datasets respectively.
3. We designed an unbiased dataset of proteins likely not forming condensates (negative dataset) based on protein-protein interaction networks, that does not necessitate any other *a priori* knowledge and assumption about the properties of the negative and positive data. Our approach is thus superior to that of PSAP¹, which used PDB as a negative dataset assuming disorder as the main determinant of condensate proteins, which is not necessarily the case. And DeePhase and FuzDrop treated all non-positive data as negative that is again a problematic approach since many yet undiscovered condensate-forming proteins are in their negative datasets.

Although PICNIC was trained on the richest human data, it generalizes well to other organisms tested. Proteome-wide predictions by PICNIC estimate that ~40% of proteins partition into condensates across different organisms, from bacteria to humans, with no apparent correlation with organismal complexity or disordered protein content. Thus surprisingly, while disorder content of a proteome correlates with organismal complexity, condensate protein content does not.

We believe that not only are method but also our analysis across organisms will spark interest in the scientific community.

Regarding the Reviewers' comment about which features drive condensation we showed the feature importance about the synuclein family above, and we repeat here our comment to Reviewer 1:

“Extracting the most important features for each individual protein of the algorithm can give a hint about properties, specific for this particular protein, that are important for its condensation. Here is the example for FUS protein where the most important features are listed (New Supplementary Figure S12). The features highlighted by PICNIC for FUS protein condensation are in agreement with a study which shows that phase separation is primarily governed by multivalent interactions among tyrosine residues from prion-like domains and arginine residues from RNA-binding domains, which are modulated by negatively charged residues. Glycine residues enhance the fluidity, whereas glutamine and serine residues promote hardening (Wang, J. et al, 2018). Thus, analysing PICNIC features can generate hypothesis for potential mutations that may alter condensation.

New Supplementary Figure S12. The top features picked by PICNIC to predict FUS (Uniprot ID: FUS_HUMAN) condensation.

Response to Reviewers' comments

Reviewer #1 (Remarks to the Author):

I appreciate the additional work the authors performed by introducing additional metrics, analysis and experiments. Together with the changes to text narrative, I think these make the paper more interesting and more robust.

Thank you for the positive comment.

However, there is one crucial point I disagree on. It is simply not true that the standard practice to validate predictors is to only test the positive predictions. If one wanted to use any metric to evaluate accuracy of PICNIC, such as ROC AUC, PRCs, Matthews correlation coefficient, or even a simple confusion matrix, one would need to know the true negative rate. Given the relatively simple set-up of their validation experiment, I don't see why one would avoid this.

We agree that knowing the predictive performance on true negative data is crucial. Nevertheless, testing negative predictions is not so commonly done, since negative experiments are harder to prove.

This seems to me even more important in this context, where the authors state that 40-50% of any proteome is predicted to form condensates. To test the accuracy of the method, but also to highlight whether it would truly be a useful tool, it is crucial to know what the background rate of condensation is. I understand it's hard to find a gold standard set of negatives but to claim a conceptual advance, one set of negatives tested in the same conditions as the positives is needed. If, in the literature, a set of sequences tested in similar conditions exists, then that would also be ok.

Indeed, there is no gold standard dataset and very few reported examples of condensate negative proteins. In a recent paper by Hou et al. (<https://www.nature.com/articles/s41467-024-46445-y>), 3 negative predictions were tested experimentally. All those 3 proteins also score negative according to our algorithm providing further experimental validation for PICNIC see table below. Please note, that one of this proteins, GTF2A2 forms condensates in our experiments (details below and in Response Figure 2).

Uniprot ID	Gene name	PICNIC score
Q9Y2D0	CA5B	0.41
Q8NE65	ZNF738	0.11
P52657	GTF2A2	0.36

If, as the authors state in their rebuttal text, they don't see a point in testing negatives as these sequences could anyway condense under specific circumstance, then what is the point of this - or any other - condensation predictor and what is the meaning of the current validation experiments (performed in just one specific set of conditions)?

We are sorry for the confusion. We did not mean that testing negatives has no point, we meant that testing positives gives usually a high confidence result, while showing negative experiments does not prove that the proteins would be negative in all conditions. Testing negative predictions under exact same experimental conditions as positives has definitely value.

Inspired by the reviewer we started to systematically explore our negative predictions and also some previously published negative data. We randomly selected 21 proteins predicted to be negative by

PICNIC after filtering out the ones that are associated with membranes since these are “obvious negatives” (Response Dataset 1).

We expressed the fluorescently tagged proteins in U2OS cells to visualize them and noticed, that 5 of them do not express at all (despite repeated attempts) and one localized to the ER. The remaining 15 proteins also showed weak expression. Overall, out of these 15 proteins, 10 formed no condensates, 2 formed low confidence condensates and 3 formed condensate-like foci (Response Figure 1).

c

		PICNIC PREDICTIONS		
		Condensate	No condensate	Accuracy
EXPERIMENTS	High-confidence condensate	18	3	86%
	Low-confidence condensate	3	2	
	No condensate	3	10	77%
	Total number	24	15	

Response Figure 1. Most (10 out of 15) proteins predicted not to form condensates indeed do not form detectable foci.

a) Representative images of the U2OS cells expressing the tested proteins tagged with a fluorescent protein (RFP). All images are scaled to the scale bar 10 micron (shown as yellow bars). We found 10 out of the 15 tested proteins did not form condensates, while two proteins GTF2A2 and PABIR3 formed low confidence condensates in the minority of cells (yellow square), and 3 proteins formed high confidence condensates (blue squares).

b) Wide range of structural motifs covered in the test proteins; AlphaFold2 structural models of the proteins are colored according to secondary structures. Many of the tested proteins have large IDRs and yet do not form condensates. **c)** Summary of the experimental validation including 24 positive and 15 negative predictions. Using the strict threshold of assuming all low-confidence condensates are also positive, 67% of the negative predictions were correct, while assuming less strict threshold, 77% of the negative predictions were correct. The accuracy of identifying positive predictions is 86%, which is in line with our benchmarking on test datasets as shown in Figure 2.

Curiously, the protein previously reported to form no condensates in a recent paper by Hou et al. (see answer above), also formed low confidence condensates in our experiments (GTF2A2), which would give the method of Hou et al. a negative predictive value of 66%, inferior to ours.

Response Figure 2. Previously reported condensate negative protein, GTF2A2 expressed in HeLa cells tagged with GFP (left panel, Figure 5b middle from Hou et al. Nat Comm) form low-confidence condensates in our experiments (right panel) expressed in U2OS cells and tagged with iRFP.

We were puzzled by the findings about the *conditional* condensation of GTF2A2 and decided to perform a systematic assessment of condensate formation of the entire dataset of proteins (24 positive and 21 negative predictions) and recorded a qualitative assessment of their expression level as well as condensate formation on their own and in the presence of four/five condensate markers to test localization to known condensates. The results are summarized in Response Dataset 1 (attached as an excel file). [The Response Dataset 1 has been removed from the peer review file.]

These large-scale experiments revealed the following findings.

- The predicted negative proteins showed lower expression and toxicity. Specifically, 6 out of 21 were not expressed on their own (ND means no expression). While all proteins predicted to be positives expressed well in *all* conditions tested.
- We observed that some of the predicted negative proteins form *conditional* condensates, i.e. they form condensates when co-expressed with certain condensate markers. 17 proteins had good expression when co-expressed with stress granule marker FMR1 (see as an example Response Figure 3). We hypothesize that their toxicity is buffered by sequestration into condensates, such as stress granules.

- In contrast, the positive predictions robustly form condensates across cells and perturbations (such as co-expression). While the negative predictions do not form condensates or only form condensates in few conditions in few cells.

[Response Figure 3 has been removed from the peer review file.]

In our view, from a philosophical perspective, there are no “true negatives”, i.e. proteins that *never* form condensates. There may be an *in vivo* condition for every protein under which it forms *conditional* condensates. However, it is clear from these systematic experiments, that PICNIC correctly identifies proteins that are less likely to form condensates, but cannot predict if they *never* form condensates.

These findings inspired us to add a section on the limitations of our model and the limitations of the data that is available to the scientific community, and impact all other computational tools equally. We think, these are very important points and strengthen our conclusions (see below excerpts from the Discussion).

We are grateful to the reviewer for advocating for testing negative predictions in our experiments. These turned out to be a very complex, and therefore we would like to avoid presenting Response Figure 1 in the current paper. We are preparing a follow up paper about this important topic. Specifically, Response dataset 1 forms the basis for a new manuscript that assesses systematic conditional condensation and what are the most important parameters to record in order to facilitate computational predictors with complex outcomes (such as toxicity, expression level, condensation in certain conditions). We are in the process of validating our hypotheses about the toxicity and sequestration into condensates as a response.

In our view, presenting in the current manuscript the *simplistic view* of single experiment-based annotation of *condensate negative proteins* would not be beneficial for the scientific community. Specifically, it would be misleading, since some of the proteins that would be classified negative from these experiments would form *conditional* condensates based on our co-expression experiments.

Our data argue against a dichotomic view of condensate formation, in favour of a focus towards the specific conditions that induce condensation, which will also require a call for action to increase the efforts from the community to collect more complete data recording for the precise experimental conditions. These efforts would benefit the development of computational predictors in the future. Therefore, we would like to include here data only on the positive experiments and the extensive

discussion on the caveats of the model and the data regarding negative predictions (see Discussion below).

“The algorithms in this paper are designed to understand whether a protein has the potential to localize to a condensate, trained on previous experimental data. We tested 24 predicted proteins and found that 87.5% form foci *in cellulo*, and 75% form high confidence condensates based on a quantitative definition of condensates. In making this analysis, we set a cut-off above the diffraction limit (Figure S9). This does not mean that clusters of proteins under this limit do not form condensates, but that super-resolution techniques would be required to analyze this. Three proteins formed no condensates, nevertheless we cannot be certain that they would not form condensates under different conditions or cell-types.

Further, the number of cells with condensates can be highly variable. Such noise is common in biology, as often phenotypes are not fully penetrant. Even isogenic populations of cells and organisms show phenotypic variability, e.g. due to transcriptional noise⁵⁶. We see the same phenomena with condensate formation, and therefore introduced the stringent criteria that high-confidence condensate forming proteins have foci in *most* imaged cells, while low-confidence condensates form foci in *few* cells only. We showed that several of the tested proteins localize into known condensates, P-bodies or to the nucleolus (Figure 5). It is important to state that the condensate localization makes no claim to biological function in cells. These condensates were observed in cell lines, using iRFP-tagged proteins that had varied expression levels. It is possible that some proteins will only form condensates under stress or in certain cell types at certain concentrations. In some cases, a protein for instance might not localize to a condensate under physiological conditions, but might when overexpressed in cancer. Therefore, although each protein can localize to a condensate, it remains to be sorted out by detailed experiments whether any individual protein localizes to a condensate in a specific cell type at a certain concentration and using different tags and/or antibodies.

PICNIC successfully categorized three proteins (ZNF738, GTF2A2, CA5B) as negatives. These proteins were recently shown not to form condensates, as part of a concerted effort to assess the efficacy of computational predictors⁵⁷. However, proving that a protein never forms a condensate is philosophically impossible, since this can be cell-type and condition-specific. This conundrum hinders current machine learning based efforts to train classifiers. We argue against a dichotomic view of condensate formation, in favour of a focus towards how likely condensation occurs under specific conditions. We assume that the proteins that are predicted to form condensates by PICNIC would robustly form condensates across cells (e.g. high confidence condensates). On the other hand, the proteins that are predicted *not* to form condensates should not form condensates in *most* conditions. In

order to escape the puzzle of unobtainable negative data, we anticipate that future generations of condensate predictors will not just limit themselves to simple yes/no classification of condensate proteins, but will also incorporate condition-specific condensation.”

Reviewer #2 (Remarks to the Author):

The authors adequately addressed my concerns.

We are glad to hear that, thank you!

Reviewer #3 (Remarks to the Author):

The authors have partially enhanced their manuscript, incorporating the database and software package as previously suggested. However, they have not fully addressed some of the concerns raised in earlier comments.

1. The synuclein section's feature importance lacks clarity on whether the features positively or negatively impact predictions. They should consider utilizing SHAP values for clarity, and marking significant feature locations on the protein structure.

These are great suggestions. We have already shown the absolute value of SHAP values (called feature importance in the old plot) and now we also indicate residues on the protein structure that contribute to the top significant features, which corroborate that the bending of the alpha-helix may contribute to the correct predictions for beta-synuclein.

Supplementary Figure S14. The top features picked by PICNIC regarding the synuclein protein family: alpha, beta and gamma-synuclein (threshold for $\text{abs}(\text{SHAP values})$ displayed 0.07, 0.08, 0.08, respectively). Negative SHAP values mean the feature is present, while positive values mean that the feature is absent in a given protein. There are many shared features that contribute to the prediction for all three synucleins (grey). The structure-based features (red) were mapped on the respective 3D models (AlphaFold2) of the proteins.

2. I remain skeptical about the predictive results in a minority of species being conclusive for the tree of life. The supposed irrelevance might stem from a lack of observed correlation in a limited set of species. I advise more caution in describing the results in Fig.6, avoiding overgeneralized conclusions.

We computed proteome-wide PICNIC scores from another 10 organisms (25 in total). The lack of correlation in old Figure 6 remained (see new Figure 6, $R^2=0.08$). This increased and more representative set gives us higher confidence in our observation and conclusion. Nevertheless, we toned down the conclusion, see text:

“Interestingly, while the fraction of disordered proteins increases with organismal complexity as shown before^{49,50}, we found no correlation between fraction of predicted condensate proteins in a proteome and the disordered protein content (**Figure 6c**) across the 25 species tested even when using different metrics to assess the disorder of a proteome (Figure S12).”

Figure 6. Inferring condensate proteins across the tree of life reveals no correlation with disorder content.

a) PICNIC model is species-independent. We validated the PICNIC model on known condensate proteins from 14 different species (defined by CD-CODE). PICNIC correctly identified 70-100% of known condensate proteins of all species tested, except for zebrafish (50%). **b) Proteome-wide prediction of proteins in biomolecular condensates by PICNIC predictor.** **c) Disorder content and fraction of condensate-forming proteins of a proteome are not correlated.** The fraction of disordered proteins (proteomes shows no correlation across 25 selected organisms from bacteria, archaea to mammals (Pearson $R^2 = 0.08$).

Response to Reviewers' comments

Reviewer #1 (Remarks to the Author):

I appreciate the experimental effort the authors undertook and I am glad it has contributed to opening the way for a follow up paper. The lack of “true” negatives should definitely make us all move away from a binary view of condensation. It should also help to define their condition-specific role in cellular fitness, a topic which over the years has been touched upon in different publications with evidence for both condensation-induced cellular toxicity or condensation-induced fitness advantages.

For transparency I recommend including the tested negative set at least in the supplementary file.

I am ok with the rest of the revisions to text and figures and with publication of the manuscript.

Answer: Thank you, we are also grateful for the thoughtful comments.

We agree to include the list of negatives and we added the imaging of those proteins as Figure S11 and extended the discussion on finding “true negatives”.

Reviewer #3 (Remarks to the Author):

The authors have resolved my concerns, and I agree to accept this article.

Answer: Thank you!